# High Salinity Shelf Water production rates in Terra Nova Bay, Ross Sea from high-resolution salinity observations

Una Kim Miller [1] ✉, Christopher J. Zappa [1], Arnold L. Gordon [1],
Seung-Tae Yoon [2], Craig Stevens [3,4] & Won Sang Lee [5]

High Salinity Shelf Water (HSSW) formed in the Ross Sea of Antarctica is a precursor to Antarctic Bottom Water (AABW), a water mass that constitutes the bottom limb of the global overturning circulation. HSSW production rates are poorly constrained, as in-situ observations are scarce. Here, we present high-vertical-and-temporal-resolution salinity time series collected in austral winter 2017 from a mooring in Terra Nova Bay (TNB), one of two major sites of HSSW production in the Ross Sea. We calculate an annual-average HSSW production rate of ~0.4 $Sv$ ($10^6\ m^3\ s^{-1}$), which we use to ground truth additional estimates across 2012–2021 made from parametrized net surface heat fluxes. We find sub-seasonal and interannual variability on the order of 0.1 $Sv$, with a strong dependence on variability in open-water area that suggests a sensitivity of TNB HSSW production rates to changes in the local wind regime and off-shore sea ice pack.

High Salinity Shelf Water (HSSW) is a key component of the ~5.4 Sverdrups ($Sv$; $10^6\ m^3\ s^{-1}$) of dense shelf water (DSW) that flow off the continental shelves of Antarctica to form Antarctic Bottom Water (AABW), a water mass that constitutes the bottom limb of the Meridional Overturning Circulation (MOC) and ventilates the deep ocean[1–3]. HSSW is produced in coastal polynyas, which are formed when intense katabatic winds, originating from the Antarctic Ice Sheet and channeled seaward through coastal valleys, continually push newly formed sea ice offshore. Polynyas serve as windows through which a wintertime ocean, otherwise sealed off by ice, exchanges gas, heat, and momentum with the atmosphere. Intensive heat loss to the atmosphere from these polynyas makes them highly productive sea ice factories[4] and the resulting brine plumes feed reservoirs of HSSW at depth. HSSW in the Ross Sea is of particular interest because substantial changes in its salinity observed over the past six decades have directly influenced the properties of AABW in the Pacific and Indian sectors of the Southern Ocean. From the 1960's onward, Ross Sea HSSW has freshened at a rate of $\sim 0.03\ decade^{-1}$[5,6], resulting in the freshening of newly formed AABW sourced from DSW exiting the

western Ross Sea[5,7,8] and contributing to a weakening of the bottom limb of the MOC[2,9,10]. This freshening is associated with the impacts of climate change on the broader Western Antarctic region, wherein a warming ocean has increased the amount of glacial meltwater advected into the Ross Sea from the nearby Amundsen and Bellingshausen Seas[6]. Shorterterm variability in Ross Sea salinity has been tied to climate anomalies, such as the co-occurrence of a positive Southern Annular Mode (SAM) with extreme El Niño conditions from 2015-2018 that resulted in a sudden reversal of the decades-long Ross Sea freshening trend[11,12]. This anomaly induced weaker easterly winds in the Amundsen Sea, reducing sea ice import into the Ross Sea and allowing for an increase in local ice production, and thus brine rejection and salinity, across the Ross Sea continental shelf[13]. A subsequent recovery of AABW salinity downstream of the western Ross Sea outflow followed[13,14]. However, the long-term freshening trend suggests that if such reversals are not persistent, Ross Sea HSSW and the AABW it feeds will drop below their defining densities by mid-century[6]. As the MOC is thought to regulate Earth's climate on centennial to glacial-interglacial timescales[9,15], variability in AABW has implications for our

[1]Lamont-Doherty Earth Observatory of Columbia University, Palisades, New York, USA. [2]Kyungpook National University, Daegu, South Korea. [3]National Institute of Water and Atmospheric Research, Wellington, New Zealand. [4]University of Auckland, Auckland, New Zealand. [5]Korea Polar Research Institute, Incheon, South Korea. ✉e-mail: ukm2103@columbia.edu

presently warming climate system and places great importance on our understanding of HSSW formation.

HSSW in the Ross Sea is formed primarily within the Terra Nova Bay (TNB) and Ross Sea Polynyas[16], which occur regularly every austral winter. It is broadly defined by temperatures near freezing and salinities greater than 34.62, though HSSW formed in TNB, bounded by Nansen Ice Shelf to the west, Cape Washington to the north, and Drygalski Ice Tongue to the south[17,18] (Fig. 1), is uniquely salty and dense, with salinities exceeding 34.8[16,19]. Therefore, despite a smaller total volume contribution (33%)[20], TNB has an outsized impact on the density of overall Ross Sea HSSW. Furthermore, it directly supplies much of the estimated $0.2-0.54$ $Sv$ of HSSW exported from the western Ross Sea via the Drygalski Trough[20-23]. HSSW is produced only during austral winter, once salinification of surface waters via brine rejection has broken down lingering stratification from summer ice melt and solar radiation, allowing full water column convection to occur and brine plumes to reach depth[24-26]. While some of the HSSW produced in TNB is converted to Ice Shelf Water (ISW) through its interaction with the local glaciers[23] or exported southward under Drygalski Ice Tongue[12,20], most of it spreads northward through the Drygalski Basin, taking about 8 months to travel through the Drygalski Trough and reach the continental shelf break[27]. It ultimately exits the continental shelf as dense, tidally-modulated gravity plumes[21,22,27-29]. Though the circulation of HSSW within TNB and its export via the Drygalski Trough are somewhat understood[12,23], its production rate is poorly constrained; previous estimates using parameterized net surface heat fluxes[30] and simulated passive tracer experiments in a high-resolution regional model of the Ross Sea[20] differ by ~0.9 $Sv$, a discrepancy on the same order of magnitude as the estimated transport of TNB HSSW northward to the shelf break. More tightly constrained estimates of HSSW production rates in TNB are needed to better understand its variability and contribution to DSW exported from the western Ross Sea.

In this study, we present estimates of HSSW production rates made using moored high-spatial-and-temporal-resolution salinity measurements collected during austral winter 2017 within the TNB polynya. The mooring was deployed from February 2017 through February 2018 at 74.97° S, 163.96° E, approximately 7 $km$ east of the Nansen Ice Shelf in a region of TNB that best exemplifies the response of the polynya to the regional katabatic regime, dominated by drainage through Reeves Glacier[31] (Fig. 1). The number of sensors and the mooring's relatively shallow location on a local bathymetric high in 390 $m$ water depth allowed for an unusually high density of measurements across the full water column in a region of TNB central to polynya growth, ice formation, and HSSW production. Moored instrumentation utilized in this study include 7 temperature-salinity sensors at depths ranging from 47 $m$ to 360 $m$ and an Acoustic Wave and Current Profiler (AWAC) with Acoustic Surface Tracking (AST) capabilities deployed at 37 $m$ facing upward at the sea surface (Methods). The primary goal of this paper is to utilize data from a uniquely positioned and densely instrumented mooring to calculate HSSW production rates in TNB from continuous, in-situ observations. We use these estimates to explore several aspects of HSSW production, including the conversion of HSSW into Ice Shelf Water (ISW) and its variability in relation to the katabatic wind regime, as well as to ground-truth a method for estimating production rates from parameterized net surface heat fluxes in order to examine potential inter-annual variability in the context of broader Ross Sea salinity trends.

## Results and discussion
### HSSW production rates from moored salinity time series
Production of HSSW begins in early July, once turbulent mixing induced by katabatic winds breaks down stratification and allows brine rejection at the surface to salinize subsurface waters[24]. We define this break down of stratification and the start of the HSSW production season to begin once the water column at the mooring site is mixed,

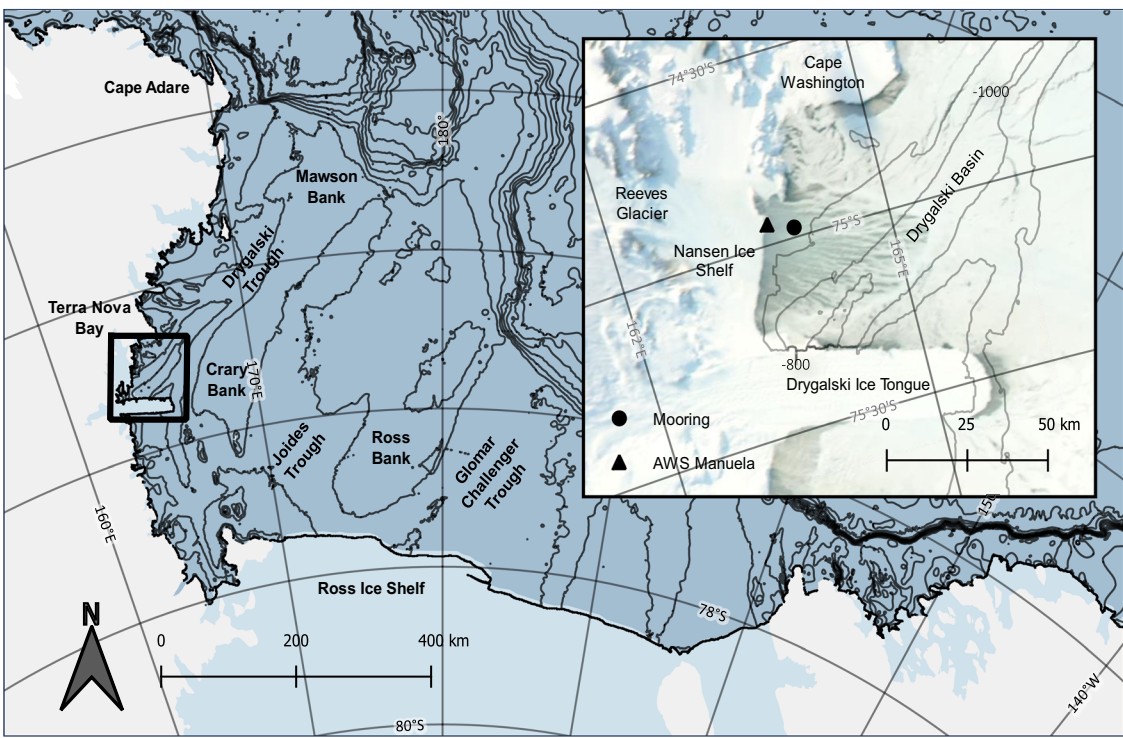

**Fig. 1 | The Ross Sea with an inset map of Terra Nova Bay (TNB).** In the inset, visible imagery (NASA Worldview) of the open TNB Polynya is shown overlain with the locations of the mooring (74.97° S, 163.96° E) and Automatic Weather Station (AWS) Manuela (74.92° S, 163.6°). Bathymetric contours (General Bathymetric Chart of the Oceans; GEBCO) are shown at 250 $m$ intervals in the larger map and 200 $m$ in the inset. The Ross Sea basemap was obtained from the Quantarctica mapping environment[99].

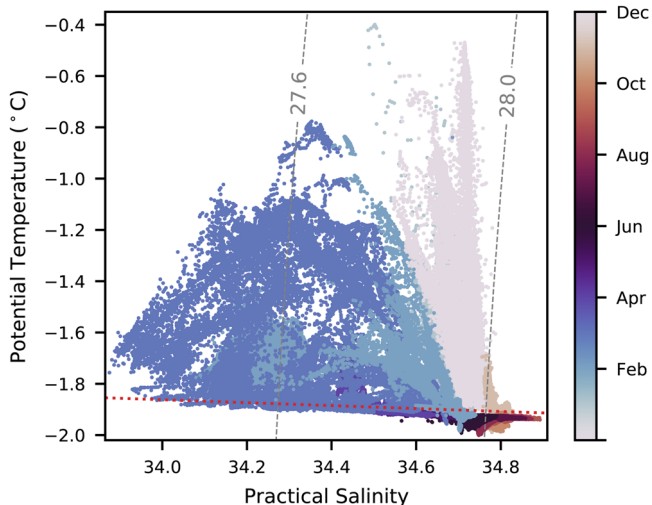

**Fig. 2 | A temperature-salinity diagram from the moored sensor at 47 $m$ depth showing the presence of High Salinity Shelf Water (HSSW) during the austral winter months.** Grey dashed lines denote potential density contours ($\sigma_0$ [$kg\,m^{-3}$]) and the red dashed line marks the salinity-dependent surface freezing temperature. HSSW is defined by $\sigma_0 \geq 28\ kg\,m^{-3}$.

i.e., potential density at the deepest instrument (360 $m$) is within 0.03 $kg\,m^{-3}$ of that at the shallowest sensor[32]. HSSW production events were systematically identified (Methods) where increases in salinity resulted in potential densities at the 47 $m$ instrument of or exceeding 1028 $kg\,m^{-3}$, the defining density threshold of TNB HSSW[12,20] (Fig. 2). Events were discarded if salinity increases appeared to be associated with processes other than in-situ ice production, determined by manually checking that each event was part of an expected chain of events associated with the opening of the polynya: first, an increase in wind speeds measured by Automatic Weather Station (AWS) Manuela to excess of 25 $ms^{-1}$, marking the start of a katabatic wind event (Fig. 3a), followed by AST returns showing a transition from hard reflectors (>-70 $dB$; thick sea ice) at the sea surface to soft reflectors (<-60 $dB$; air bubbles, sea spray, and frazil ice) scattered above and below the surface (Fig. 3b), indicating the movement of sea ice offshore and an opening of the polynya, and finally, an average current velocity in the offshore direction (Fig. 3d) (Methods). In most all events, average current velocities measured by the upward-facing AWAC were uniformly eastward in the direction of the winds and showed little variability in speed, with standard deviations of order 0.01 $m\,s^{-1}$. However, some events were discarded because of strong shoreward currents or abrupt changes in current direction that were clearly decoupled from the westerly katabatic winds and possibly associated with eddy activity previously observed in TNB[33,34]. HSSW may have still been produced during these discarded events, but the brine rejection signal was obscured and thus the events were unsuitable for our analysis.

We identified 27 HSSW production events over the course of austral winter of 2017, with production beginning in July and continuing through October. The integrated increase in salinity across all 7 instruments at the start of each event was used to calculate rates of brine rejection (Methods). We note that the relative changes in salinity at each depth tracked one another and indicated that brine rejected at the surface was mixed to depth, as in Fig. 3c. The resulting brine rejection rates, also referred to as salt fluxes, at the mooring site were extrapolated across the full area of the polynya, derived from a daily 3.125 $km$-resolution sea ice concentration (SIC) product, to estimate HSSW production rates (Methods). The average production rate across the 27 events is 3.76 $Sv$, with a 95% bootstrapping confidence interval of [2.89, 4.75] (Methods; Supplementary Table 1). Conversion to an annual average yields 0.43 $Sv$ [0.34, 0.55] (Methods).

The event-averaged ice production rate calculated from the measured salt fluxes is 67 cm day$^{-1}$ (Methods, Supplementary Table 1), a value consistent with estimates of ice production rates in TNB from observations collected on a field campaign that occurred in May 2017[35,36], several months into our mooring deployment.

## Sub-seasonal variability in HSSW production tied to the katabatic wind regime

The major components of the HSSW production rate calculation, polynya area and salt flux, are shown in Fig. 4b, and the resulting HSSW production rates are shown duplicated in Fig. 4c–e, colored by the average strength, average duration, and frequency of katabatic wind events, respectively, in the week preceding each production event. The same katabatic wind characteristics calculated across the 2 and 3 weeks prior to each production event exhibit similar patterns and are not shown here. Katabatic winds events were identified as periods of time during which westerly (from 225–315°) winds measured at AWS Manuela exceeded 25 $ms^{-1}$ for more than 1 hour[24]. Katabatic wind event frequency in units of $week^{-1}$ was calculated as a cumulative count of wind events over the 1-week period. Variability in the calculation of HSSW production (Fig. 4c–e) is clearly driven by that of its polynya area component (Fig. 4b; Eq. 3), with pronounced peaks in polynya area, and therefore HSSW production rates, occurring on August 08 and September 18. These peaks appear associated with high frequencies (Fig. 4e), strengths (Fig. 4d), and durations (Fig. 4ce) of preceding katabatic winds events. This is consistent with previous mooring-based studies that have shown interannual variability in the salinity of HSSW at depth[12,24] and polynya area[37] in TNB to be linked to katabatic wind event frequency and duration. An additional mechanic may be in effect, however: it has been previously suggested that the thickness, and therefore ease-of-advection, of ice at the periphery of a polynya can modulate the response of its growth to winds[31,38,39]. We speculate that this is why the peak in polynya area and HSSW production on August 08 corresponds to anomalously frequent, but neither especially strong nor long-lasting, katabatic wind events in the week prior. Frequent clearing of sea ice would have resulted in a prevalence of newly formed, and more easily advected, thin ice, allowing for greater polynya area expansion in response to ensuing katabatic winds of even relatively mild strength and duration. Likewise, the final, late-season peak on October 30 may have resulted despite relatively low katabatic wind event strengths, durations, and frequencies in the week prior because months of polynya openings acted to diminish the overall presence of thick ice. This mechanic would also explain why relatively high strengths and durations of katabatic wind events preceding the production event on July 18 did not result in a peak in polynya area and HSSW production; the low frequency of katabatic wind events may have allowed pack ice to consolidate in the week prior, lowering the responsiveness of polynya growth to the wind regime.

## Conversion of HSSW into TISW

Most of the annual-average ~0.4 $Sv$ of HSSW that we estimate to be formed in TNB are likely transported northward towards the continental slope[20,21,27,40], with some unknown portion of the remainder converted into Ice Shelf Water (ISW). ISW is a water mass that forms when HSSW is circulated beneath an ice shelf, and due to the depth-dependence of freezing point, melts the basal ice[41]. Mixing of HSSW with the resulting meltwater produces ISW that either sinks along the continental shelf to directly contribute to the exported DSW[21,42] or rises along the front of the ice shelf as a buoyant plume[43,44]. In TNB, HSSW is cooled and diluted through interactions with Nansen Ice Shelf and Drygalski Ice Tongue, forming a distinct water mass known as TNB ISW (TISW)[12,23,35,45], defined by temperatures below surface freezing point (~−1.95° C)[16] and potential density below that of the defining threshold of HSSW, 1028 $kg\,m^{-3}$[12]. TISW serves as a cold end-member

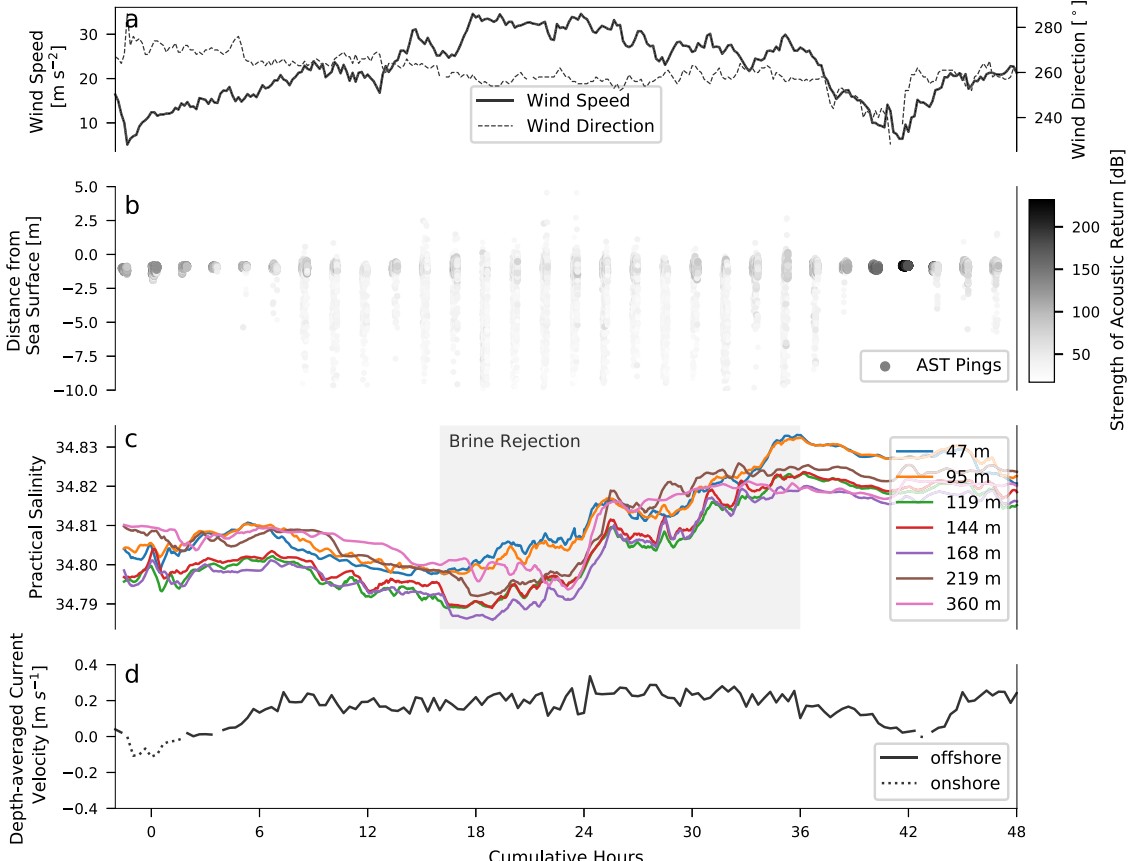

**Fig. 3 | An example High Salinity Shelf Water (HSSW) production event used to calculate production rate.** Shown are (**a**) wind speed and direction from Automatic Weather Station (AWS) Manuela, (**b**) acoustic surface tracking (AST) returns, where hard reflectors (e.g. conglomerated ice) are shown in darker shades of gray and soft reflectors (e.g. sea spray, injected air bubbles, suspended frazil ice) are shown in lighter shades of gray, (**c**) 30-min rolling averages of salinity at each of the 7 sensors with the identified brine rejection signal shaded in gray, and (**d**) current velocities averaged over the upper -35 *m* of the water column.

for DSW exported off of Cape Adare and while its volumetric contribution is thought to be small, it may contribute significantly to the sinking of DSW off the continental shelf due to the thermobaric effect[21]. We can attempt a simple budget analysis to constrain its volume rate of production: Gordon et al.[21] estimated that an annual average of -0.2 *Sv* of HSSW exit the western Ross Sea, based on hydrographic surveys collected during the Anslope-2 program in February-April 2004, which showed 0.8 *Sv* of benthic-layer water transported past Cape Adare, with approximately 25% (-0.2 *Sv*) consisting of HSSW with salinity >34.70. If approximately 0.3–0.5 *Sv* of HSSW are formed in TNB, based on the 95% confidence interval calculated for the average of our mooring-based estimates, and approximately 0.2 *Sv* exit the western Ross Sea[21], then assuming minimal transport south of Drygalski Ice Tongue[20], we can speculate that up to $\mathcal{O}(0.1Sv)$ may be converted to TISW. This residual represents an upper bound on TISW, as it includes all possible pathways for HSSW other than direct transport northward to the continental shelf, which could include conversion to a lower salinity HSSW through mixing with surrounding water masses in and out of TNB, or even non-negligible transport southward. There is also considerable uncertainty in this budget analysis that arises from the assumption that the export of HSSW past Cape Adare, estimated based on data from 2004[21], is constant at 0.2 *Sv*. As we explore in a later section, HSSW production rates in TNB may vary on the order of 0.1 *Sv* year to year. An alternate calculation of TISW production rate from satellite-based estimates of basal meltwater flux from Nansen Ice Shelf and Drygalski Ice Tongue suggests TISW to be an order of magnitude smaller: total basal meltwater flux is reported to be between 3.5 and 8.7 *Gt year*$^{-1}$ [46,47] in the recent decade, which, assuming a meltwater density of 1000 *kg m*$^{-3}$, is

approximately equivalent to 0.0001–0.0002 *Sv*. Further assumption that TISW volume is composed of approximately 0.3% glacial meltwater yields an TISW production rate $\mathcal{O}(0.01Sv)$ (Methods).

**Revisiting prior estimates of HSSW production rate**

Previously reported estimates of HSSW production rate in TNB range from 0.28 *Sv*[20] to 1.2 *Sv*[30], a discrepancy of ~0.9 *Sv* and source of error that is on par with the annual-average ~0.8 *Sv* of DSW reported exiting the Western Ross Sea past Cape Adare[21]. The rate of 0.28 *Sv* was reported by Jendersie et al.[20], based on simulated passive tracer experiments in a high-resolution coupled ocean-ice shelf model of the Ross Sea. It falls only slightly lower than the 95% confidence interval of our 2017 estimate (Table 1) and sources of uncertainty, such as the use of Special Sensor Microwave/Imager (SSM/I)-derived surface heat and freshwater fluxes in lieu of explicit ice production, would likely produce confidence intervals that overlap significantly with our own. The latter study, Fusco et al.[30], estimated production rates from surface heat flux parameterizations calculated using reanalysis over the period of 1990-2006. This calculation, first published by Van Woert et al.[48], assumes a balance between net surface heat loss from TNB and latent heat gain from the production of sea ice, allowing production rates of HSSW to be inferred from the resulting estimates of sea ice production rates. A constant polynya area of 1300 *km*$^2$, based on satellite imagery from 1979[17], is assumed. The resulting average of 1.2 *Sv* is significantly higher than the our average value of -0.4 *Sv*, as well as that of Jendersie et al.[20]. Interannual variability in HSSW production may contribute substantially to this discrepancy with our 2017 value: a calculation of HSSW production rate anomalies following the methods described in Fusco et al.[30] and using European Center for Medium-Range Weather

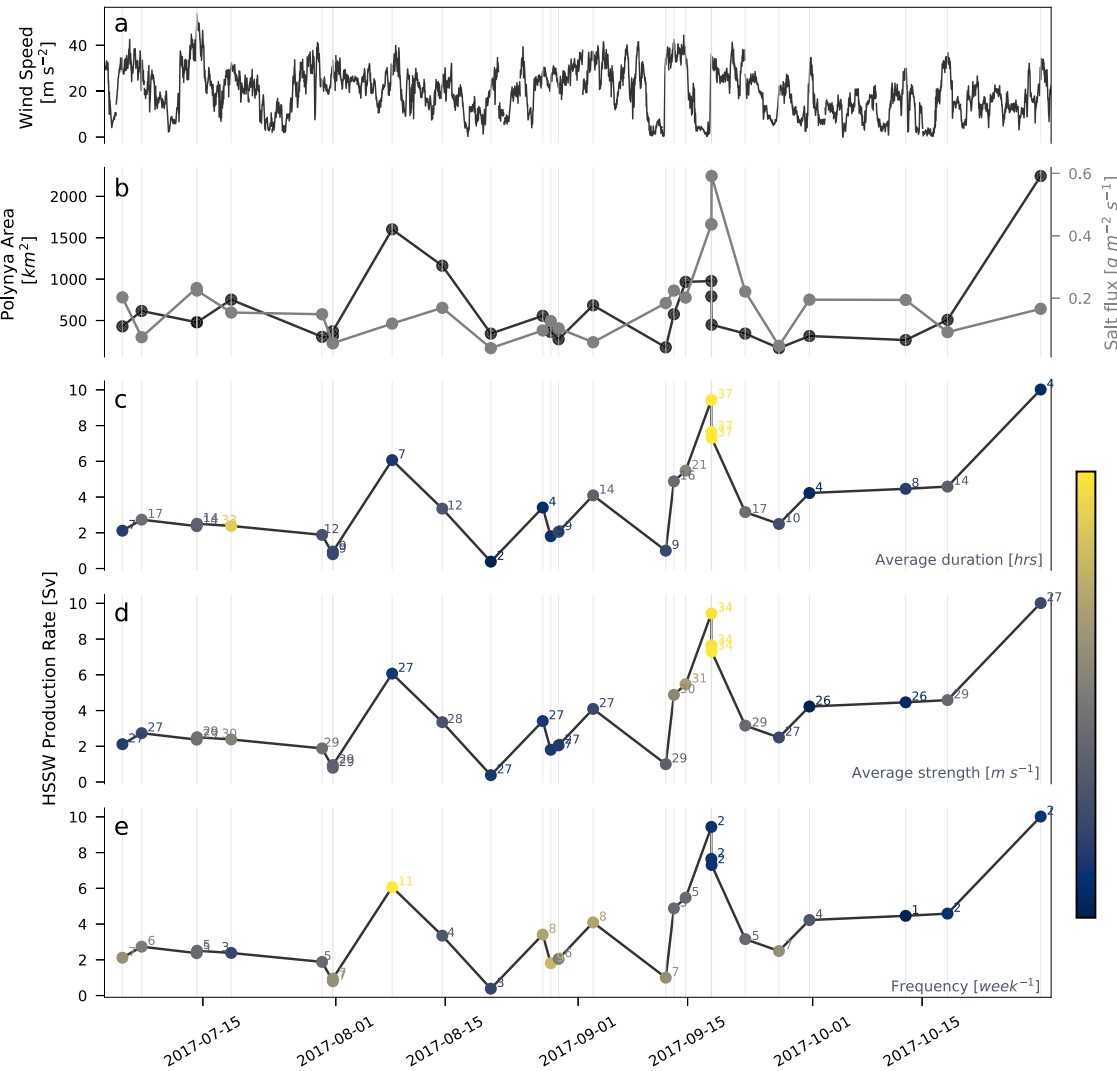

**Fig. 4 | High Salinity Shelf Water (HSSW) production rates across austral winter 2017, the major components of their calculation, and the connection between production rates and the katabatic wind regime.** Shown are (**a**) wind speeds at Automatic Weather Station (AWS) Manuela, (**b**) polynya area and salt fluxes (brine rejection rates) calculated for each HSSW production event, and the resulting HSSW production rates colored by average katabatic wind event (**c**) duration, (**d**) strength, and (**e**) frequency in the week preceding each HSSW production event. Numbers to the right of the data points in **c**–**e** denote the magnitude of each katabatic wind event statistic, with colors normalized to the respective lowest (blue) and highest (yellow) values. Source data for **b**–**e** are given in Supplementary Table 1.

**Table 1 | Estimates of annual-average HSSW production rates in TNB with reported confidence intervals (CI) and standard deviations (STD)**

|  | Data | Study period | TNB HSSW production rate |
|---|---|---|---|
| Present Study | In-situ salinity measurements | July – October 2017 | 0.43 Sv;95% CI [0.34,0.55] |
| Present Study | Parameterized net surface heat fluxes | 2012-2021 | 0.38 *and* 0.74 *Sv*; *STD* [0.10 *and* 0.20] |
| Fusco et al. (2009) | Parameterized net surface heat fluxes | 1990 – 2006 | 1.2 *Sv*; STD [0.3] |
| Jendersie et al. 2018 | Simulated passive tracer experiments | N/A | 0.28 *Sv* |

The two values given for the heat-flux based estimates from the present study were calculated using heat transfer coefficients of $1.1 \times 10^{-3}$ and $2.2 \times 10^{-3}$, respectively.

Forecasts (ECMWF) Reanalysis Version 5 (ERA5) from 1990–2021 suggests the average difference between the 1990–2006 period and 2017 to be ~0.5 *Sv*. Uncertainties inherent to the parameterization of heat fluxes and other key components of the calculation likely also contribute, however. As in-situ wind data from AWS Manuela did not become consistently available until 2012, surface heat fluxes in Fusco et al.[30] were parameterized using wind speeds from ERA 40-Year (ERA40), which, like many reanalysis products,

underestimates wind speed magnitudes and dynamics in Antarctica[49-55]. To account for this, Fusco et al.[30] applied a heuristic correction to their calculation of HSSW production rates based on a comparison of heat flux-based estimates calculated using available AWS wind speeds against those using reanalysis wind speeds in a single year[50]. Also notable is the sensitivity of the calculation of turbulent heat fluxes to the choice of heat transfer coefficients, for which a scarcity of in-situ data has limited the development and validation of in

polar environments[56]. Fusco et al.[30] follows the heat flux parameterizations of Budillon et al.[56], using coefficient values of $C_H = C_E = 1.75 \times 10^3$ derived from eddy correlation measurements of heat fluxes made from masts mounted onto thick sea ice cover in the Arctic[57]. However, the suitability of a range of coefficients, from values closer to those typically used in open-ocean settings (e.g. $1.1 \times 10^3$ [58]) to as high as $2.2 \times 10^3$ [36], can also be argued (Methods). Lastly, our examination of sub-seasonal variability in HSSW production rates showed a strong dependence on polynya area; the application by Fusco et al.[30] of the same polynya area across all 16 years of the study period would not capture year-to-year variability in average polynya size, which satellite observations show to be on the order of hundreds of square kilometers for the TNB Polynya[31,37].

## Interannual variability in HSSW production from parameterized heat fluxes

Despite the uncertainties associated with the use of parametrized heat fluxes to estimate HSSW production rates, it remains a necessary approach in a region where in-situ data collection is sparse. We use this approach to estimate HSSW production rates across 2012-2021, allowing us to place our mooring-based estimate from austral winter 2017 into a broader context. We make several changes to the calculation implemented by Fusco et al.[30], discussed in the previous section and further detailed in Methods, which include: (1) Use of in-situ wind speed measurements from AWS Manuela in place of those from reanalysis, (2) calculation of average polynya area specific to each year of the study using an Advanced Microwave Scanning Radiometer 2 (AMSR-2) SIC product ground-truthed against visible satellite imagery, (3) calculation of net surface heat fluxes only across the months of active HSSW formation (July through October[24]) rather than the entire year, and (4) use of low and high heat transfer coefficients of $C_E = C_H = 1.1 \times 10^{-3}$ and $2.2 \times 10^{-3}$, respectively, to bound uncertainty due to the choice of coefficient. We find that this modified calculation yields annual-average HSSW production rate values for 2017 bounded by 0.38 $Sv$ (using the lower transfer coefficient) and 0.74 $Sv$ (higher coefficient) in austral winter 2017, overlapping with the confidence interval of our mooring-based rate and suggesting that this approach may be used to estimate HSSW production rate with some accuracy. Coincidentally, these values are the same as the averages of values calculated across the full 2012-2021 period, which we report in Table 1. Regardless of the absolute values of the estimated production rates, however, we believe this method to be a useful tool in assessing potential relative changes in production rate over time.

HSSW production rates calculated from 2012−2021 are shown in Fig. 5, which indicates that our mooring-based estimate in 2017 may be part of a broader trend of increasing production rates in the latter half of the decade. The primary components of this calculation are net surface heat flux out of TNB and polynya area (Eqs. 12, 13). As with the sub-seasonal mooring-based estimates, variability is largely driven by polynya area, with the increase in production rates from 2015 onward clearly reflecting an increase in year-to-year average polynya area (Supplemental Figure 1). Net surface heat flux actually decreases during this period due to a decreasing sensible heat component, suggesting that increases in ice production rates did not contribute to the apparent increase in HSSW production rates. Based on the connection we found between polynya area and katabatic wind strength, duration, and frequency on a sub-seasonal scale, we might expect the same mechanics to explain the increase in polynya area, and therefore HSSW production rates, from 2015 onward. However, while a peak in average wintertime katabatic wind speeds and total hours of katabatic winds in 2017 is reflected in a slightly higher average production rate in the same year, there is no clear interannual trend in katabatic wind conditions concurrent with the increase in polynya area. Instead, we speculate that the increasing polynya area in TNB was enabled by the same process that Silvano et al.[13] found to facilitate the recovery of western Ross Sea HSSW salinity: a decrease in sea ice import from the Amundsen Sea to the Ross Sea beginning in 2014 that resulted in lower sea ice concentrations across the Ross Sea continental shelf, allowing for greater sea ice divergence and open-water areas (and therefore, in-situ sea ice production). These same conditions may have made it easier for the TNB Polynya to expand to increasingly larger average areas, despite little interannual change in the katabatic wind regime.

## Implications

Variability in the properties and production rate of AABW impacts the strength of MOC, and thus the capacity of the ocean to sequester atmospheric heat and carbon and replenish abyssal oxygen[9,59,60]. Though the recently observed recovery of the salinity, density, and thickness of AABW downstream of the western Ross Sea was driven by conditions favoring increased sea ice production across the broader Ross Sea continental shelf rather than local processes within TNB[13], this does not preclude the latter from contributing to AABW variability. As TNB HSSW is the densest component of DSW exported from the Ross Sea, substantial changes in its rate of production could feasibly alter the proportion of HSSW within, and thereby the density of, downstream AABW. Our estimates of annual-average HSSW production rates show variability of $\mathcal{O}(0.1)$ $Sv$ sub-seasonally as well as interannually. On both time scales, this variability is driven by changes in polynya area of $\mathcal{O}(100)$ $km^2$. Across austral winter 2017, we found changes in polynya area to be tied to katabatic wind speeds, frequency, and duration, but speculate it to be additionally modulated by the thickness of offshore ice. This same influence of offshore ice (or relative lack thereof) is possibly why the multi-year increase in polynya area, and thus estimated HSSW production rates, from 2015 to 2021

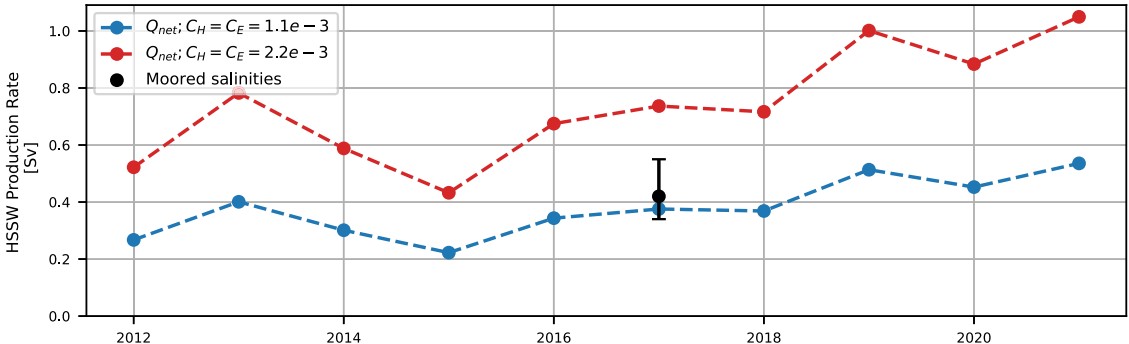

**Fig. 5 | Interannual variability in Terra Nova Bay (TNB) High Salinity Shelf Water (HSSW) production rates from 2012–2021.** The 2017 annual-average mooring-based estimate of 0.43 $Sv$ is shown in black with error bars denoting the bootstrap 95% confidence interval of 0.34−0.55 $Sv$. Annual-average rates across the decade from parametrized net surface heat fluxes calculated using turbulent heat flux coefficients of $1.1 \times 10^{-3}$ and $2.2 \times 10^{-3}$ are shown in blue and red, respectively.

occurred despite no concurrent trends in the katabatic wind regime. Therefore, the dominance of polynya area variability in the HSSW production rate calculation suggests that processes affecting either the katabatic wind regime or sea ice divergence in the TNB region have the potential to greatly impact local production of HSSW. The proportionality of increases in HSSW production rates to increases in the open-water area of the polynya must be further explored, however. Our calculations necessarily assume a constant brine rejection rate across the entire open-water area of the polynya (Eqs. 3, 13), but a katabatic wind parcel will both lose momentum and gain heat as it travels across water, reducing the magnitude of ocean-to-atmosphere heat loss at longer fetches. Reduced HSSW production rates offshore would result in a lower sensitivity to changes in polynya area than our calculations imply. We expect our conservative definition of polynya area as the open-water area defined by SIC less than 30% (Methods) to limit the significance of spatial variability in our calculations, but further mooring-based measurements of brine rejection rates across the open-water area of TNB are needed to confirm this. Additional mooring deployments would also help to address a key question that arises from our work: how does the variability we have observed in TNB HSSW production rates manifests downstream in DSW and ultimately, AABW? Specifically, co-deployed moorings in TNB and the Drygalski Trough would allow for direct observation of the magnitude and time scale at which local variability in HSSW production propagates. In light of the changes recently observed in AABW, ongoing, in-situ monitoring of its critical source regions is necessary to constrain the future of MOC.

## Methods

### Dataset

Moored instrumentation utilized by this study consisted of 7 SeaBird Electronics conductivity and temperature (SBE37-SM/SMP MicroCAT) sensors at depths of 47, 95, 119, 144, 168, 219, and 360 $m$ and an upward-facing 600 kHz Nortek Acoustic Wave and Current (AWAC) profiler at 37 $m$ depth, collecting both current velocity profiles in the upper ~35 $m$ of the water column as well as Acoustic Surface Tracking (AST) measurements. The 47 and 95 $m$ MicroCATs were pumped and sampled at a rate of once every 2 min while the remaining MicroCATs were unpumped and sampled once per minute. Temperature and salinity measurements were validated against a shipboard conductivity-temperature-depth (CTD) cast performed upon deployment of the mooring. Temperature and salinity at 47 $m$ and 95 $m$ were additionally validated against a CTD cast performed upon recovery of the mooring; deeper MicroCats had ceased sampling in the weeks prior. The AWAC profiler head has four beams, three at an angle that collect velocity profiles and a fourth vertical beam in the center, on which Acoustic Surface Tracking (AST) relies. AST ensonifies the surface of the ocean and records the distance from the instrument head that corresponds to the strongest decibel return. It is often used to calculate wave statistics and under certain conditions may be used to measure the thickness of ice pack[61], but here we use it to more generally assess the presence of ice at the surface. The AST footprint at the surface depends on its depth of deployment, with a 40 $m$ depth corresponding to a surface footprint 1.2 m in diameter[62].

Measurements of air temperature, humidity, wind speed and direction, and atmospheric pressure collected by Automatic Weather Station (AWS) Manuela (74.97° S, 163.93° E) are made available by the University of Wisconsin-Madison AWS Program. Data were downloaded at 10-minute resolution for the years of 2012-2021, with 2012 being the first full year of data available. Additionally, shortwave and longwave radiation measurements collected at the nearby AWS Rita (74.72° S, 164.03° E) in 2009 were obtained for the purpose of ground truthing the parameterizations of shortwave and longwave heat fluxes.

Single-level European Center for Medium-Range Weather Forecasts (ECMWF) Re-Analysis 5 (ERA5) parameters across the years of 2012 through 2021 were obtained from the Climate Data Store at a 6-hour timestep and 0.25° × 0.25° spatial grid. ERA5 proceeded ERA-interim in 2017, and features higher spatial and temporal resolution, among other improvements[63]. Variables selected were 2-meter air temperature, 2-meter dew point, sea surface temperature, land-sea-mask, mean sea level pressure, and total cloud cover. Data were subset to an area representative of Terra Nova Bay (TNB), a box bounded by 75.3°S, 74.66°S, 163.5°E, and 167.0°E. Any non-ocean data points within this geographic subset were masked by removing data where land-sea-mask had a value greater than 0.5.

A daily 3.125 $km$-resolution sea ice concentration (SIC) product is made available by the University of Bremen, generated from Advanced Microwave Scanning Radiometer 2 (AMSR-2) imagery using the Arctic Radiation and Turbulence Interaction Study (ARTIST) Sea Ice (ASI) algorithm. The ASI algorithm exploits the difference in horizontally and vertically polarized brightness temperatures at high frequencies (86.5 GHz) between open water and sea ice[64] to assign concentration percentages ranging from 0–100 for each pixel. SIC maps were obtained for the year of 2012-2021. Visible imagery collected by the National Aeronautics and Space Administration (NASA) Aqua, Terra, and Suomi satellites were obtained across the year 2017 from NASA Worldview for the purpose of ground-truthing SIC concentration maps.

### Calculation of polynya area

Polynya area was calculated as the sum of SIC pixels representing open water multiplied by the area of each pixel (3.125$^2$ $km^2$). The value of SIC that delineates the open water and thin ice of the polynya from surrounding sea ice cover is called a threshold value and is heuristically[31] assigned values ranging from 15%[65,66] to upwards of 70%[67-70], depending on the product and its spatial resolution. To determine a suitable threshold value for the ASI SIC product used in this study, we compared visual satellite imagery of the open polynya in March and April 2017 with corresponding SIC scenes from the same days. The Suomi, Aqua, and Terra satellites pass over TNB at approximately 04:00, 05:00 and 20:00. Because the polynya is known to vary in size on scales as short as hours[71,72], only days in which the polynya area remained the same across all three flyover periods were used to ground truth the SIC product, which represents a daily-averaged view. Two distinct regions of the open polynya are apparent in the visible imagery: a dark blue open-water area marked by grease or shuga ice streaks aligned with the offshore katabatic winds and an apparent thin sea ice area bordered by a ridge of thicker accreted ice (Supplemental Fig. 2). The open-water area defined by wind streaks is almost certainly actively forming new ice, and previous studies have used these streaks as a visual check on the extent of the TNB polynya[37,68]. It is unclear what stage of ice formation the thin ice area represents, as it may be actively forming new ice with HSSW production rates comparable to those of the open-water area, or it may be a conglomerated mass of ice such as pancake ice, where air-sea interaction and therefore in-situ ice and HSSW formation would be reduced. These regions correspond to SIC threshold values of roughly 30% and 60%, respectively. We select a threshold of 30%, which is consistent with the wind-streak definition of prior studies and close to the 20% threshold used by a recent study using the same SIC product in TNB[12].

The University of Bremen SIC product uses an outdated land mask that does not account for the recent calving of the Nansen Ice Shelf in April of 2016. To correct for this, SIC maps in 2017 were overlaid on the visual satellite imagery showing the present boundary of the Nansen Ice Shelf and open-water pixels were manually added in to fill the gap wherever *SIC* was less than 30%. The average number of pixels added

across July-October 2017 was 25. This average of 25 pixels was added to calculations of polynya area in all years beginning in 2016.

### Calculation of HSSW production rate from in-situ salinity measurements

High Salinity Shelf Water (HSSW) production rates were calculated from individual production events, which were identified from the moored time series as follows:

1. First, periods of time were flagged during which the potential density time series, calculated from salinity and temperature at the 47 $m$ instrument, exceeded the defining threshold of HSSW ($1028\,kg\,m^{-3}$)[12].
2. These time periods were divided into distinct HSSW production events, defined to start/end when potential density at the instrument first increased above/decreased below $1028\,kg\,m^{-3}$, with the interval between start and end being at least 1 hour.
3. To filter out increases in salinity due to processes other than in-situ brine rejection (e.g., tidal or eddy-related advection), events were manually checked for association with active katabatic winds, as determined by wind speeds at AWS Manuela, an opening of the polynya as determined by scattered AST returns at the surface, and currents in the direction of the wind in the upper 35 $m$ as measured by the AWAC.

Katabatic winds here are defined as nominal wind speeds at AWS Manuela greater than $25\,m\,s^{-1}$ [24] and coming from a direction of $225 - 315°$, as in Yoon et al.[12]. The defining magnitude of katabatic winds is somewhat subjective, with some studies defining katabatic winds less strictly, using lower magnitude threshold values (e.g., $17\,m\,s^{-1}$ [35]) or wind speeds referenced to 10 $m$ rather than nominal wind speeds. However, all of the identified HSSW production rate events either directly followed or were concurrent with katabatic winds as defined by wind speeds greater than $25\,ms^{-1}$, thus none were discarded as a result of this criterion.

The above steps to identify and filter HSSW production events yield 27 distinct events occurring from early July through October, each of which is used to calculate a HSSW production rate. First, the mass of salt in units of $g\,m^{-2}$ at the mooring site is calculated as

$$m_s(t) = \int_0^H \rho\,S\,dz \tag{1}$$

$H$ is the depth to which salinity changes can be attributed to in-situ brine rejection[73], which at the wintertime-mixed mooring site is designated as the depth of the deepest sensor, 360 $m$. The potential density of sea water ($\rho$) in $kg\,m^{-3}$ and practical salinity, $S$, assigned units of $g\,kg^{-1}$, are calculated using the Gibbs Seawater Toolbox[74].

The rate of brine rejection in units of $g\,m^{-2}\,s^{-1}$ at the mooring site is then calculated as

$$\frac{dm_s}{dt} = \frac{m_s(t_1) - m_s(t_0)}{\Delta t} \tag{2}$$

across an interval of time defined by an increase in salinity due to brine rejection that begins at time $t_0$ and ends at time $t_1$ (e.g. gray-shaded region in Fig. 3c). Alternatively, $\frac{dm_s}{dt}$ is taken to be the slope of the least-squares linear fit of the data between $t_0$ and $t_1$. We opt for the latter, as it is less sensitive to high-frequency variation in the rate of brine rejection as well as to the somewhat subjective choice of $t_0$ and $t_1$. For two separate events, density remained elevated above $1028\,kg\,m^{-3}$ through the following day, suggesting a steady-state balance between brine rejection and advection offshore. An additional production rate was calculated for the following day using the initial brine rejection rate with the second-day polynya area.

To estimate the rate of brine rejection across the entire polynya, $P_s$, in units of $kg\,s^{-1}$, Eq. 2 is multiplied by the area of the polynya, $A_p$, in units of $m^2$.

$$P_s = \frac{dm_s}{dt}A_p \tag{3}$$

Finally, the rate of transformation of ambient water into HSSW, or the production of HSSW, is given by

$$P_{HSSW} = \frac{P_s}{\rho_{HSSW}(S_{HSSW} - S_{LSSW})} \times 10^{-6} \tag{4}$$

in units of $Sv$. Prior studies[30,48,50] have assigned fixed, historical values to the density of HSSW ($\rho_{HSSW}$), salinity of HSSW ($S_{HSSW}$), and the salinity of the ambient water, or low salinity shelf water ($S_{LSSW}$)[29,47,49] converted into HSSW. The value of ($S_{HSSW} - S_{LSSW}$) can have a large impact on the final value of $P_{HSSW}$ while being relatively subjective to varying salinity definitions of the two water masses. This is avoided in our event-wise calculations, as our in-situ salinity measurements allow us to define $\rho_{HSSW}$ and $S_{HSSW}$ as the density and salinity of the water at the end of the HSSW production event and $S_{LSSW}$ as the salinity at the start of the event.

Values are converted to an annual average by multiplication of the ratio of cumulative days of katabatic winds from July through October, 42.12 days, to the total days in a year, 365 days. This is based on an assumption, supported by measured wind speeds, AST returns, and salinity, that whenever katabatic winds blow during the winter HSSW production season, the polynya will open and the ocean will lose heat and form new ice, producing HSSW.

### Ice production rate calculation

Ice production rates ($P_i$) in $m\,s^{-1}$ are calculated from measured brine rejection rates (Eq. 2) as:

$$P_i = \frac{dm}{dt}(s_i\rho_i)^{-1} \tag{5}$$

where $s_i$ is the salinity of frazil ice in $g\,kg^{-1}$, approximated as 0.31 multiplied by the salinity of seawater[75], and $\rho_i$ is the density of ice ($0.95 \times 10^3\,kg\,m^{-3}$).

### ISW meltwater fraction estimate

Given that Ice Shelf Water (ISW) is the product of mixing between HSSW and glacial meltwater, the fraction of meltwater can be calculated as follows:

$$Meltwater\,Fraction_{temperature} = \frac{T_{HSSW} - T_{ISW}}{T_{HSSW} - T_{meltwater}} \tag{6}$$

$$Meltwater\,Fraction_{salinity} = \frac{S_{HSSW} - S_{ISW}}{S_{HSSW} - S_{meltwater}} \tag{7}$$

With assigned values of 34.85, -1.91, 34.73, and -2.1 for $S_{HSSW}$, $T_{HSSW}$, $S_{ISW}$, and $T_{ISW}$, respectively[12] and 0 and -90.75 for $S_{meltwater}$ and $T_{meltwater}$, respectively[76], Eq. 5 yields 2.1 permille and Eq. 6 yields 3.4 permille. An average of the two values suggests that the meltwater fraction of ISW is -3 permille, or 0.003.

### Calculation of surface heat fluxes

Total surface heat flux, $Q_{net}$, is calculated as $Q_{net} = Q_s + Q_b + Q_S + Q_L$, with fluxes out of the ocean defined as negative by convention. Each component of the net surface heat flux is defined below.

**Incoming shortwave radiation.** Incoming solar radiation is estimated as

$$Q_s = (1 - \alpha)C_c T_r S_a \cos\eta \tag{8a}$$

where $\alpha$ is the albedo of the water surface, 0.08[56,77], for conditions of no ice cover, $C_c$ is a cloud cover correction, $T_r$ is the transmittance ratio of a clear sky atmosphere, $S_a$ is the solar constant, 1353 $W\ m^{-2}$, and $\cos\eta$ is the cosine of the zenith angle of the Sun. The cloud cover correction is calculated as

$$C_c = \alpha_1 + \alpha_2 C^{\alpha_3} \tag{8b}$$

where $\alpha_1 = 1$, $\alpha_2 = 0.6$, $\alpha_3 = 3$ and $C$ is total cloud cover from ERA5 reanalysis[78]. Shine[79] introduced a Zillman-type[80] model for the transmittance ratio of a clear sky atmosphere modified for polar regions,

$$T_r = \frac{\cos\eta}{\beta_1 \cos\eta + (\beta_2 + \cos\eta)e(T_d)\beta_3 + \beta_4} \tag{8c}$$

where $\beta_1 = 1$, $\beta_2 = 1$, $\beta_3 = 10^{-5}$, $\beta_4 = 0.046$ and $e(T_d)$ is the water vapor pressure in Pascals calculated from ERA5 2-meter dew point temperature, $T_d$, in Kelvin[81]:

$$e(T_d) = 611 \times 10^{7.5(T_d - 273.16)/(T_d - 35.86)} \tag{8d}$$

The cosine of the zenith angle of the sun is calculated following Marcus et al.[52]:

$$\cos\eta = \sin\phi \sin\delta + \cos\phi \cos\delta \cos h \tag{8e}$$

Values of $\cos\eta < 0$ (indicating the sun is below the horizon) are masked and the corresponding calculations of $Q_s$ are set to 0. Latitude, $\phi$, is set to $-75°$ for TNB. The solar inclination angle ($\delta$) is given by

$$\delta = 23.44° \cos(172 - d) \tag{8f}$$

where $d$ is the day of year. The solar hour ($h$) is

$$h = (12 - t_s) \times 15 \tag{8g}$$

where $t_s$ is local solar time, which is a corrected local time given by

$$t_s = local\ time + TC/60 \tag{8h}$$

TC is a correction factor:

$$TC = 4(longitude - LSTM) + EoT \tag{8i}$$

In TNB, the longitude is 165°E and the local standard time meridian (LSTM) is 195°E. The Equation of Time (EoT) is an empirical equation that corrects for the eccentricity of the Earth's orbit and the Earth's axial tilt[82].

$$EoT = 9.87 \sin 2B - 7.53 \cos B - 1.5 \sin B \tag{8j}$$

where $B = 360/365(d - 81)$.

**Net longwave radiation.** Net longwave radiation flux is calculated as

$$Q_B = 4\varepsilon\sigma T_A^3(T_A - SST) + \varepsilon\sigma T_A^4(0.39 - 0.05\sqrt{e(T_d)})(1 - \chi C^2) \tag{9}$$

Here, $\varepsilon$ is open ocean emissivity, $\sigma$ is the Stephan-Boltzman constant ($5.67 \times 10^{-8}\ W\ m^{-2}\ K^{-4}$), $T_A$ is ERA5 2-meter air temperature in Kelvin, SST is ERA5 sea surface temperature in Kelvin, and $\chi$ accounts for changes in cloud type with latitude, ranging from 0.5 at

the equator and 1 at the poles. We select a value of 0.82 for Terra Nova Bay based on Table 9 in Budyko[83] for the latitude of 75°S. Shortwave and longwave radiation as calculated from Eqs. 5 and 6 for the year 2009 show good agreement with concurrent in-situ measurements by AWS Rita (Supplemental Fig. 3).

**Turbulent heat fluxes.** Sensible heat flux is given by

$$Q_H = \rho_a c_P C_H |\bar{V}|(T_A - SST) \tag{10}$$

where $\rho_a$ is the density of dry air, $c_P$ is the specific heat of dry air (1004 $J\ kg^{-1}$ Huschke[84]), $C_H$ is the transfer coefficient for sensible heat, discussed below, $|\bar{V}|$ is the magnitude of the winds measured by AWS Manuela. Wind speeds are divided by 1.5, a correction factor based on measurements along the katabatic wind pathway in TNB[85] that aims to account for the attenuation that occurs across the distance between the weather station and the open water of the polynya.

Latent heat flux is given by

$$Q_E = \rho_a L_E C_E |\bar{V}|(q_A - q_s) \tag{11a}$$

where $L_E$ is the latent heat of vaporization ($2.5 \times 10^6 J\ kg^{-1}$; Haltiner & Martin[86]), $C_E$ is the transfer coefficient for latent heat, discussed below, which is considered to be roughly the same as $C_H$[58,87,88], and $q_a$ and $q_s$ are the water vapor mixing ratios of the air at 10 m and at the sea surface, respectively:

$$q_a = \frac{0.622e}{p_s - 0.378e} \tag{11b}$$

$$q_s = \frac{0.622e_s}{p_s - 0.378e_s} \tag{11c}$$

Here, $e$ is vapor pressure and $e_s$ is saturation vapor pressure in Pascals. The surface pressure, $p_s$, is taken to be the ERA5 mean sea level pressure in Pascals.

$$e = 611 \times 10^{a(T_d - 273.16)/(T_d - b)} \tag{11d}$$

where $(a, b) = (9.5, 7.66)$ if an ice cover exists and $(a, b) = (7.5, 35.86)$ if it does not[81]. No ice cover is assumed for the open-water area of the polynya. For the calculation of $e_s$, SST is substituted for $T_d$.

The choice of heat transfer coefficients is non-trivial, with a difference of $1 \times 10^{-3}$ resulting in differences in net heat flux of hundreds of $W\ m^{-2}$. Though functions of wind speed and atmospheric stability[89], the coefficients are commonly assigned constant values, a practice supported by early measurements in open-ocean settings that showed little variation across wind speeds[58,90,91]. Maximum wind speeds assessed in these studies were 18–25 $m\ s^{-1}$, with $C_E \approx C_H \approx 1.1 \times 10^{-3}$ in unstable atmospheric conditions. One study, utilizing a large set of flux measurements across wind speeds ranging from 5–20 $m\ s^{-1}$, found $C_E$ (with $C_H$ approximately equivalent to and calculated as a function thereof) to increase steadily from $1.08 \times 10^{-3}$ at wind speeds of 5 $m\ s^{-1}$ to $1.2 \times 10^{-3}$ at 18 $m\ s^{-1}$, suggesting a slight dependence on wind speed[89]. Model runs of Coupled Ocean–Atmosphere Response Experiment (COARE) Algorithm[89,92], a widely-used bulk flux algorithm that computes the coefficients as functions of atmospheric stability, extrapolated this result to $\sim 1.3 \times 10^{-3}$ at 25 $m\ s^{-1}$[89]. However, little observational data exist in the kinds of extreme conditions occurring in TNB, where katabatic winds regularly reach strengths upwards of 40 $m\ s^{-1}$. Field and lab studies in hurricane conditions show the drag coefficient, $C_D$, to level off at winds speeds between 30–40 $m\ s^{-1}$, likely due to the effects of wave breaking and the generation of a sea foam layer at the ocean surface[93,94]. Laboratory experiments show the enthalpy flux coefficient, related to $C_E$, to level off and even decrease at

speeds greater than 20 $m\ s^{-1}$ [95]. Based on the observed and inferred behavior of the heat transfer coefficients at winds ranging from 5-40 $m\ s^{-1}$, it would not be unreasonable to apply a transfer coefficient close to the typical open-ocean value of $1.1 \times 10^{-3}$ in a polynya setting. This would be consistent with coefficients obtained using an exponential, fetch-dependent transfer coefficient formula developed from flux measurements near arctic leads[96], which yields -1.1 × 10⁻³ at longer fetches on the order of hundreds of meters (such as across a polynya)[97]. However, a recently published study of shipboard bulk-flux and integral-heat-flux measurements in katabatic wind conditions in TNB suggests sea spray to greatly enhance sensible heat fluxes, resulting in a transfer coefficient of $\sim 2.2 \times 10^{-3}$. This higher value is closer to the value used by Fusco et al.[30] and others[56] of $1.75 \times 10^{3}$, which was originally derived from eddy correlation measurements of heat fluxes made from masts mounted onto thick sea ice cover in the Arctic[57], as well as with the value of $2.0 \times 10^{3}$ used in a widely cited polynya modeling paper[98]. We assign $C_E = C_H$, opting to use low and high values of $1.1 \times 10^{-3}$ and $2.2 \times 10^{-3}$, respectively, as a means of bounding our heat flux estimates.

### Calculation of HSSW production rate from net heat fluxes

The rate of ice production, $P_i$, is defined[98] as

$$P_i = Q_{net}/L_f\rho_i \qquad (12)$$

where $L_f$ is the latent heat of fusion ($3.34 \times 10^5\ J\ kg^{-1}$) and $\rho_i$ is the density of ice ($0.95 \times 10^3\ kg\ m^{-3}$). We calculate $P_i$ from $Q_{net}$ parameterized across the years 2012-2021 (Equations 8-11) during the months of active HSSW production, July through October. The rate of brine rejection across the polynya (corresponding to Eq. 3 in the mooring-based calculation) is calculated as

$$P_s = \rho_i P_i A_P (s_w - s_i) \qquad (13)$$

where $A_P$ is calculated using the ASI SIC product from July-October of each year, $s_w$ is water salinity (set to the same value as $S_{LSSW}$), and $s_i$ is the salinity of frazil ice ($0.31 s_w$)[76]. $P_{HSSW}$ is then calculated as in Eq. 4, where values of $\rho_{HSSW}$, $S_{HSSW}$ and $S_{LSSW}$ are set to 1028 $g\ kg^{-1}$, 34.81, and 34.79, respectively. The latter two values are the average $S_{LSSW}$ and $S_{HSSW}$ values across the 27 production events observed in the 2017 mooring time series. We make the assumption that the magnitude of $(S_{HSSW} - S_{LSSW})$, 0.03, in Eq. 4 stays constant across the 10 years over which we use parameterized heat fluxes to infer interannual HSSW production rate variability, but acknowledge that this introduces further uncertainty to these estimates; changes to $(S_{HSSW} - S_{LSSW})$ on the order of 0.01 result in changes to $P_{HSSW}$ on the order of 0.01 $Sv$. Production rates are converted to an annual average via multiplication by the number of months over which net heat fluxes were calculated, 4, and the total months in a year, 12.

### Data availability

AWS Manuela data are available from https://amrc.ssec.wisc.edu/aws/index.php?region=Reeves%20Glacier&year=2017&mode=uw, the shortwave and longwave radiation measurements used to ground truth our parameterized fluxes is available from Columbia University Academic Commons via https://academiccommons.columbia.edu/doi/10.7916/D8805F2P, single-level European Center for Medium-Range Weather Forecasts (ECMWF) Re-Analysis 5 (ERA5) data are available from the Climate Data Store via https://cds.climate.copernicus.eu/cdsapp#!/dataset/reanalysis-era5-single-levels?tab=overview, the daily 3.125 $km$-resolution SIC is available from the University of Bremen via https://seaice.uni-bremen.de/data/, and visible imagery from the Aqua, Terra, and Suomi satellites is available from NASA Worldview (https://worldview.earthdata.nasa.gov/). Source data (polynya areas, brine rejection rates, HSSW production rates, and katabatic wind statistics) for Fig. 4 are provided in Supplemental Table 1. The General Bathymetric Chart of the Oceans (GEBCO) 2019 Grid (https://doi.org/10.5285/836f016a-33be-6ddc-e053-6c86abc0788e) utilized in Fig. 1 is available from https://www.gebco.net/. Data from the mooring, including the salinity, temperature, and ADCP-based measurements used in the present study are available from the corresponding author upon request.

### Code availability

Python code written to calculate parameterized net heat fluxes out of TNB and to estimate HSSW production rates from both parameterized net surface heat fluxes and in-situ moored salinities is available on Github at https://github.com/unamiller/TNB-HSSW-Production/tree/main.

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

## Acknowledgements

We thank the scientists and ship crew involved in cruises ANA07C and ANA08C aboard the *R/V Araon*, especially Carson Witte. U.K.M. was supported by Future Investigators in National Aeronautics and Space Administration (NASA) Earth and Space Sciences (FINESST) grant 80NSSC19K1348. The deployment and recovery of the mooring was made possible with support from National Science Foundation (NSF) grant PLR 13-41688. W.S.L. was supported by the Korea Institute of Marine Science & Technology Promotion (KIMST), funded by the Ministry of Oceans and Fisheries, Korea (RS-2023-00256677; PM23020). C.S. was supported by the New Zealand Antarctic Science Platform (ANTA1801). The AWS Manuela dataset is made available courtesy of the Wisconsin-Madison Automatic Weather Station Program, funded by NSF grant 1924730. We also acknowledge the Norwegian Polar Institute's Quantarctica package, the General Bathymetric Chart of the Ocean (GEBCO) Compilation Group, and the NASA Worldview application (https://worldview.earthdata.nasa.gov/), part of the NASA Earth Observing System Data and Information System (EOSDIS). We thank Hyun A Choi for checking our heat flux calculations.

## Author contributions

C.J.Z. and A.L.G. developed the science research objectives and conceived of the mooring deployment, with ship time support facilitated by collaboration with W.S.L and KOPRI. U.K.M led the data processing. U.K.M, C.J.Z., A.L.G., S.T.Y., C.S., and W.S.L. all contributed to the data analysis. U.K.M produced the figures and drafted the manuscript. All authors provided input on the manuscript at all stages.

## Competing interests

The authors declare no competing interests.
