## [Peer Review File · Nature Communications]

Editorial Note: In their reviews of this manuscript, reviewer #3 added some comments to the manuscript file. These comments have been omitted from this Peer Review File, as their comments published below were deemed sufficient on their own.

REVIEWER COMMENTS

Reviewer #1 (Remarks to the Author):

This work focus on High Salinity Shelf Water production in Terra Nova Bay using moored, near-surface salinity (47 m), current velocities profiles (35 m to surface), and acoustic surface tracking (AST) measurements collected during austral winter 2017.

Although the main topic is very interesting, the paper does not provide important advances of significance adequate to the high-quality standards required for a publication in Nature Communications.

Moreover, the paper has also some unclear aspects (sometimes caused by incomplete reasoning), lacks of care in presentation and takes advantage of very short time series (one year).

After a careful reading of the paper, my impression is that of a potentially valuable methodological work that could be better presented in a different, more technical journal after the necessary improvement.

For these reasons I suggest rejecting the paper and encourage resubmission to a different journal.

Reviewer #2 (Remarks to the Author):

The paper estimates the HSSW production and its 10-year variability from the atmospheric measurements. The estimation allows us to infer the oceanic conditions from relatively inexpensive atmospheric measurements, which are more available than direct ocean measurements. This paper evaluate the wind derived estimate with the near-surface ocean mooring measurements, which is not an easy measurement to make, in 2017. The method verified in 2017 is extrapolated to derive the HSSW production rate in 2012-2021. I am generally in favour of publication with a few minor issues.

As the goal is to infer the HSSW production rate from the wind speed data, it is important to describe the correlation of the HSSW production rate from the mooring data to the wind speed. The correlation coefficient is presented; however, the uncertainties in the correlation coefficient is not well described. Although the p-value is quoted, the authors do not clarify and provide number of degree of freedom (dof) and its justification. I wonder if the dof considers the autocorrelation of the wind and mooring timeseries. It is also informative to present the percentage of the segments that yields the significant correlation.

The second point is in the HSSW pathways. The authors estimate 0.6 sv of the HSSW production and 0.4 sv exit the Ross Sea. The 0.2 sv loss is compared to the TISW formation from the satellite-based estimates of basal meltwater flux (line 147-151). I am not sure if this is relevant: 1) it's unclear to me that the basal meltwater ends at the surface from the Nansen Ice Shelf. With a relatively small basal melting,

does the meltwater have enough buoyancy to rise to the surface and able to mix with the HSSW? Does the HSSW driven convection reach the depth where the basal meltwater penetrates?; and 2) it seems more appropriate to consider the mixing of HSSW with the surface runoff from the Nansen Ice Shelf, which has been documented e.g.

Bell, R., Chu, W., Kingslake, J. et al. Antarctic ice shelf potentially stabilized by export of meltwater in surface river. *Nature* 544, 344–348 (2017). <https://doi.org/10.1038/nature22048>

“Supplemental Methods” seems redundant as the content is described in “Method Section” of the paper. There are no equation labels in “Supplemental Methods”, yet the document refers to Equation 19 (line 100 of 396077_0_supp_354074_rj1gdp.docx).

Line 32: sea-ice formation

Line 131: mechanism?

Line 134 – 135: it is unclear how the frequency of the katabatic wind event is calculated. Any spectrograms to confirm 6.8 per week and 5.4 per week cycles?

Line 423 and 425: missing a space between 25 and 40, and m, should be “25 m s⁻¹” and “40 m s⁻¹”.

Reviewer #3 (Remarks to the Author):

Review of High Salinity Shelf Water production in Terra Nova Bay, Ross Sea from high-resolution near-surface salinity observations

By Una Kim Miller, Christopher J. Zappa, Arnold L. Gordon, Seung Tae Yoon, Craig Stevens, Won Sang Lee

Reviewer: Keith Nicholls

The authors use a shallow mooring in Terra Nova Bay (TNB) to calculate total brine rejection in the TNB polynya through the austral winter of 2017 for all katabatic wind events. Events were identified using data from an automatic weather station located on the nearby Nansen Ice Shelf and data from the moored sensors themselves. Satellite data were used to calculate the area of the polynya. They then develop a parameterization from a combination of weather station wind data, reanalysis data, and satellite derived polynya area data to calculate a high salinity shelf water (HSSW) production time series for TNB over an extended period in order to place the results from the mooring in a temporal context.

The authors show that the variability in wind speed prior to the HSSW production events plays a significant role in HSSW production, and that the role changes in different parts of the winter. They also find their parameterized result for 2017 compares well with the result from the mooring data, lending

credibility to the multiyear time series. Their derived time series finds an increasing TNB HSSW production rate since 2015.

The ambition of improving our ability to obtain accurate estimates of HSSW production without the need for moored instruments is, I believe, a worthy one. In the main, my comments cover some areas where the choice of methodology needs more evidence in its support.

The text is nicely written, with few typographical errors that I spotted. There is an occasional curious word choice. I have made some minor textual suggestions in the attached Word document, in tracked changes. A general language point is that the word following a colon should not start with a capital letter, and for some unknown (to me) reason, ice features do not carry a definite article, so “the Nansen Ice Shelf” for example should be simply “Nansen Ice Shelf”.

I do have some more significant concerns that I would like to see addressed before this manuscript was published. I outline those below, and they are also noted in brief in the Word document.

The study has two quite separate parts: analysis of the moored sensor data, and the calculation of the HSSW production rate time series using AWS, remotely sensed, and model data. The principal connection is that the result of the mooring analysis is used to provide a degree of validation for the parameterization. It seems a shame that one of the key results from the mooring analysis (for me at least), that the HSSW productivity of a katabatic event has a significant dependence on the preceding wind variability, is not used in the parameterization. It might be that for some reason this is too difficult to implement, but it seems like a missed opportunity. It would be useful to mention that this is not being attempted here, and why that is the case. Perhaps the authors plan to implement such a scheme in the future.

The salinity and temperature data in Figure 2 (and also the example event in Figure 3) raise some important questions. Had the authors not indicated that the sensors were carefully calibrated, they would have appeared to be a bit suspect. The temperature of the majority of what is classed as HSSW is, in fact, below the surface freezing point and therefore formally ice shelf water (ISW). I also wonder if the authors can check the location of the freezing point line in Figure 2 – it looks a shade warm to me, possibly by about 0.02 degrees C? That would help bring more of the high salinity water close to the freezing point, or at it. Even then, much of the high salinity water is below even the in situ freezing point (ie taking into account the pressure at 47 m depth), which is odd and needs further comment. Having in situ supercooled water (if that’s what it is), masquerading as HSSW, is worth a mention.

When HSSW is formed at the surface, it will be at the surface freezing point, and so can only be cooled further by mixing with ISW, or by coming into contact with deep ice. I assume the authors are assuming that the former case applies here, which implies quite a lot of recycling of ISW within the polynya. If much of that ISW is recycled into HSSW the calculation of the fraction of HSSW converted to ISW (line 139 onwards) is likely to be less relevant. I think the manuscript would be improved if the authors discussed this issue. The fact that the majority of the HSSW detected is formally ISW needs a mention at least. It would be useful to include the T-S from a CTD profile, to give the data in Figure 2 some context. I recognize that they would be a summer snapshot only, but it would help the reader get their T-S

bearings.

I have a difficulty with the calculation of the HSSW production. First, equation (1) is expressed as an integral over the upper 47 m (zero to H) of the water column, except that the authors only have data from 47-m depth. It would be less confusing to express it as $H \cdot S \cdot \rho$. Eqn (1) gives the mass of salt between the surface and 47 m, and (2) shows how that salt mass is increasing during the event. Eqn (3) then integrates across the area of the polynya, and (4) converts this salt gain to an HSSW production rate. What causes me concern is that I don't understand the basis for the assumption that the increase in salinity happens only in the water column between the instrument and the surface. Is there any evidence to suggest that it doesn't descend to the sea floor? This is what I would have expected, but it's difficult to judge without knowledge of the stratification. Overall, my suspicion is that the answers from this calculation might well form a good indicator of HSSW production and its relative intensity during the events, but not an absolute value.

The authors use the assumption that the HSSW is produced only during katabatic events, and the definition of those events is highly prescriptive. I haven't tried to calculate the fraction of the period between July and October when katabatic events were active, but unless that is a very large fraction, some sort of argument is needed to convince the reader that HSSW being produced between katabatic events is insignificant. For example, if events occupied only 10% of the time then the polynya during non-katabatic conditions would need to be a very weak producer of HSSW to be making an insignificant contribution to the total.

ISW meltwater fraction estimate in line 373-380. This calculation needs to be re-written. $T_{\text{meltwater}}$ and $S_{\text{meltwater}}$ have been assigned the same values as T_{ISW} and S_{ISW} . Presumably a typo. $S_{\text{meltwater}}$ should be zero, and $T_{\text{meltwater}}$ should be roughly $T_{\text{HSSW}} - L/c$, where L is the latent heat of fusion of ice, and c is the specific heat capacity of water. This is about 86 degrees C. This then gives about the correct answers.

Lines 167+ This doesn't seem right. They are each expressed as an average for an entire year. The authors' value is also for an entire year and should not be reduced in this way. The adjustment would have been correct had they been expressed in terms of production rate per month. That would place the observed value in the middle of the published values.

Line 196 Can the authors check this. I'm not sure the correction can have been multiplicative – the units wouldn't work out. (Or am I missing something?)

Lines 348-350 I think this is the first mention of other sensors, so they come as a surprise at this stage. Might be worth giving a fuller description of the mooring earlier in the manuscript, and just saying that only the uppermost instruments will be used here (and perhaps say why).

There are several other comments in the attached marked-up version for the authors to review. I have not attempted to review the calculation of the surface heat fluxes – I leave that to a reviewer who knows more about that than me (I've also run out of time – apologies).

We sincerely thank anonymous Reviewers #1, #2 and Dr. Nicholls for their feedback. Please find our responses (yellow text) to individual comments (black text) below, and note that any line numbers are in reference to the “No/Simple Markup” version of the tracked changes manuscript document. We would like to make the reviewers aware that as part of Nature Communications software availability policy, the code showing the calculation of HSSW production rates from both moored salinity and from parameterized heat fluxes is now available at: <https://github.com/unamiller/TNB-HSSW-Production>

First, however, we would like to explain significant changes made to Figure 5, which shows the 2012-2021 time series of HSSW production rates estimated using parametrized net heat fluxes.

During the process of revising the manuscript, a colleague of ours found an error in our heat flux calculation. We miscalculated the saturation vapor pressure (e_s) used to calculate the water vapor mixing ratio at the sea surface (q_s) and ultimately, latent heat flux (Q_E). Specifically, e_s should be calculated using the “surface temperature”, and we mistakenly assigned this as ERA5 air temperature at 2 meters height (T_A) instead of sea surface temperature.

$$e_s = 611 \times 10^a (T_A - 273.16) / (T_A - b) \quad (\text{Eq. 10d})$$

$$q_s = \frac{0.622 e_s}{p_s - 0.378 e_s} \quad (\text{Eq. 10c})$$

$$Q_E = \rho_a L_E C_E |\bar{V}| (q_A - q_s) \quad (\text{Eq. 10a})$$

This correction changes the calculated latent heat fluxes, and therefore net heat fluxes, by about -100 W/m² and increases the resulting HSSW production rates. Please see the “Original” (top right) and “Fixed latent heat flux calculation” (top left) versions of Figure 5 below.

Because the resulting HSSW production rates in the “Fixed latent heat flux calculation” version extend to values significantly outside the error bars of our mooring-based calculation, we checked each aspect of the equations used to calculate production rate from net heat fluxes. These equations are:

$$P_i = Q_{net} / L_f \rho_i \quad (\text{Eq. 11})$$

$$P_s = \rho_i P_i A_P (s_w - s_i) \quad (\text{Eq. 12})$$

$$P_{HSSW} = \frac{P_s}{\rho_{HSSW} (S_{HSSW} - S_{LSSW})} \times 10^{-6} \quad (\text{Eq. 4})$$

The ($S_{HSSW} - S_{LSSW}$) term in Equation 4 has a large impact on the final value of HSSW production rate (P_{HSSW}), while being relatively subjective. In the original text (Van Woert, 1999), this term refers to the salinity change associated with the conversion of “Low Salinity Shelf Water” to HSSW, where S_{HSSW} is assigned 34.8 and S_{LSSW} is assigned 34.5. In our calculation of P_{HSSW} from moored salinity, these values

are much less subjective, as they set to the starting and ending salinity of each production event. In our calculation of P_{HSSW} using heat fluxes, these values were set to the average of the aforementioned starting and ending salinities, 34.79 and 34.82, respectively. We find that performing linear regression (see below figure) of the salinity time series over the same four months used to calculate net heat flux (July through October) and using the beginning and ending values of the regression (34.74 and 34.8) in Equation 4 lowers P_{HSSW} estimated using net heat fluxes closer to the range of the mooring-based estimate. We acknowledge that the magnitude of heat-flux-based estimates of P_{HSSW} is highly sensitive to both the choice of heat transfer coefficient (as discussed in Lines 193 - 198) and to the value of $(S_{HSSW} - S_{LSSW})$ (now discussed in Lines 398-403), but maintain that it provides valuable insight on *relative* changes to production rates across the decade. This point is now emphasized in Lines 222-224.

An unrelated, but equally impactful change, is our changing of the integration depth in Equation 1 from 47 m to 30 m in response to Dr. Nicholl’s questioning of the appropriate integration depth (see comment and response on Page 9 of this document). This lowers the HSSW production rate from an average of 0.6 Sv to 0.4 Sv, resulting in the “Final” (bottom right) version of Figure 5 shown above.

Reviewer #1:

This work focus on High Salinity Shelf Water production in Terra Nova Bay using moored, near-surface salinity (47 m), current velocities profiles (35 m to surface), and acoustic surface tracking (AST) measurements collected during austral winter 2017.

Although the main topic is very interesting, the paper does not provide important advances of significance adequate to the high-quality standards required for a publication in Nature Communications.

Moreover, the paper has also some unclear aspects (sometimes caused by incomplete reasoning), lacks of care in presentation and takes advantage of very short time series (one year).

After a careful reading of the paper, my impression is that of a potentially valuable methodological work that could be better presented in a different, more technical journal after the necessary improvement.

For these reasons I suggest rejecting the paper and encourage resubmission to a different journal.

We understand your concern regarding clarity in our manuscript, as the other two reviewers raised issues with the presentation of several major components of our analysis. These included the statistical relationship between High Salinity Shelf Water (HSSW) production rates and katabatic wind characteristics, the definition and calculation of HSSW production, the nature of Ice Shelf Water (ISW) formation and how it factors into our budget analysis for ISW production, as well as numerous other smaller, but important, points. We have made extensive revisions to the manuscript and discussed these issues at length in the present document, and hope you will find the revised manuscript clearer and more compelling.

We would like to respectfully dispute the notion that the use of a one-year timeseries in this region is a detriment to our study. Observational data are sparse in Antarctica, especially data collected in the near-surface ocean during winter, when harsh conditions prohibit ship-based activity and the keels of passing icebergs pose a real threat to upper-ocean moorings. As we assert in Lines 78-80 of the revised manuscript, this mooring provides an unpresented look at active brine rejection (resulting in HSSW formation) in the upper ocean in a polynya across an entire winter season. This uniquely allows us to estimate production rates from in-situ salt fluxes that are directly tied to the opening of the polynya and subsequent ice formation. Previous estimates of HSSW production rate have relied ice production

rates calculated from heat fluxes parameterized using reanalysis, an approach with many inherent uncertainties (discussed in Lines 172 – 207). We argue that our ability to estimate HSSW production using in-situ salt fluxes provides a valuable contribution to the Ross Sea research community's understanding of sub-seasonal variability of HSSW production in Terra Nova Bay and how it is tied to the katabatic wind regime, as well a valuable means for ground truthing the more accessible heat-flux methodology for calculating production rates.

Reviewer #2:

The paper estimates the HSSW production and its 10-year variability from the atmospheric measurements. The estimation allows us to infer the oceanic conditions from relatively inexpensive atmospheric measurements, which are more available than direct ocean measurements. This paper evaluates the wind derived estimate with the near-surface ocean mooring measurements, which is not an easy measurement to make, in 2017. The method verified in 2017 is extrapolated to derive the HSSW production rate in 2012-2021. I am generally in favour of publication with a few minor issues. As the goal is to infer the HSSW production rate from the wind speed data, it is important to describe the correlation of the HSSW production rate from the mooring data to the wind speed. The correlation coefficient is presented; however, the uncertainties in the correlation coefficient is not well described. Although the p-value is quoted, the authors do not clarify and provide number of degrees of freedom (dof) and its justification. I wonder if the dof considers the autocorrelation of the wind and mooring timeseries. It is also informative to present the percentage of the segments that yields the significant correlation.

Thank you very much for bringing the deficiencies in our statistical analysis to our attention. Upon meeting with a group of statistical consultants available through Columbia University, we have also come to understand that our use of sliding-windowed data is not suitable for regression and that the significance of our calculated regression coefficients is affected by autocorrelations in the time series (as you have pointed out). Furthermore, both regression and correlation coefficients assume normality in the data, which is not always true of our data, especially given the small ($n=10$) windowed sample sizes, and correlation assumes a lack of outliers, which is not always true for the windowed wind data.

We have decided to forgo statistical analysis on sliding-windows of HSSW production rate and katabatic wind, strength, and frequency in the one, two, and three weeks preceding each HSSW production event. Instead, we now simply plot the HSSW production rate time series colored by/labeled with katabatic wind duration, strength, and frequency in the one week preceding each event. This is shown in the updated Figure 4 and discussed in Lines 108-135. We believe that forgoing the calculation of regression or correlation coefficients simplifies the analysis and its visual representation, and more clearly demonstrating the roles that different aspects of the katabatic wind regime play in modulating HSSW production. We hope you agree with this change and that our analysis is clearer for it.

The second point is in the HSSW pathways. The authors estimate 0.6 sv of the HSSW production and 0.4 sv exit the Ross Sea. The 0.2 sv loss is compared to the TISW formation from the satellite-based estimates of basal meltwater flux (line 147-151). I am not sure if this is relevant: 1) it's unclear to me that the basal meltwater ends at the surface from the Nansen Ice Shelf. With a relatively small basal melting, does the meltwater have enough buoyancy to rise to the surface and able to mix with the HSSW? Does the HSSW driven convection reach the depth where the basal meltwater penetrates?; and

2) it seems more appropriate to consider the mixing of HSSW with the surface runoff from the Nansen Ice Shelf, which has been documented e.g.

Bell, R., Chu, W., Kingslake, J. et al. Antarctic ice shelf potentially stabilized by export of meltwater in surface river. *Nature* 544, 344–348 (2017). <https://doi.org/10.1038/nature22048>

- 1) Rusciano et al. (2013; <https://doi.org/10.1016/j.csr.2013.04.002>) gives the definition of ISW as the water mass “formed by the interaction of salty HSSW with the base of the glacial ice”. ISW is formed when HSSW at the depth of the grounding line (~200m) is circulated beneath the ice shelf, rather than through basal meltwater rising and then mixing with HSSW near the surface. When HSSW comes into contact with the basal ice, its high salinity lowers the melting point of the ice, causing it to melt. This is the source of meltwater that directly contributes to ISW formation, therefore we must consider the basal ice melt rate. Though we do not show data from deeper instruments (as they are used in the analysis of a second, forthcoming manuscript), you’re correct that HSSW convection reaches the full depth of the water column (by late June, once katabatic winds have fully broken down the lingering summertime stratification). We have added text to Lines 137-148 to improve our description of ISW formation and make the justification for using basal melt rates clearer. We realize our prior description of ISW formation processes was insufficient and thank you for bringing this to our attention.
- 2) Because ISW is defined to form directly from the interaction of HSSW with basal ice (which we hope is now made clearer in the manuscript), surface meltwater that is episodically discharged during summer months, when HSSW production is not produced, would not be a direct consideration in our budget. We do think summertime meltwater discharges of this magnitude could have an interesting and potentially large impact on water column stratification and Antarctic Surface Water formation at our mooring site, however, and will consider it in the second manuscript currently underway with this mooring dataset.

“Supplemental Methods” seems redundant as the content is described in “Method Section” of the paper. There are no equation labels in “Supplemental Methods”, yet the document refers to Equation 19 (line 100 of 396077_0_supp_354074_rj1gdp.docx).

The supplemental methods section has more details (coefficient values, equations) on the heat flux calculations, but we agree that it is not necessary to reproduce these calculations in the supplement. We have added the additional details to the original text within the manuscript and made sure all equations are numbered. The supplemental methods section has been deleted.

Line 32: sea-ice formation

Thank you for catching this. A hyphen has been added.

Line 131: mechanism?

Agreed, though this sentence has since been deleted.

Line 134 – 135: it is unclear how the frequency of the katabatic wind event is calculated. Any spectrograms to confirm 6.8 per week and 5.4 per week cycles?

We realize that was not made clear. Frequency was simply calculated as the number of events per week. When calculated over the two- and three-week periods, the number of events was normalized to 1 week, resulting in non-integer values. For example, if we count 16 events over a three-week period, we report it as $16/3 = 5.33$ events $week^{-1}$. This is now clarified in Line 118. Please note

that in simplifying this analysis in response to your valid concerns over our use of regression, we are now only considering the one-week period.

Line 423 and 425: missing a space between 25 and 40, and m, should be “25 m s⁻¹” and “40 m s⁻¹”. Thank you. This has been fixed.

Reviewer #3 (Keith Nicholls):

The authors use a shallow mooring in Terra Nova Bay (TNB) to calculate total brine rejection in the TNB polynya through the austral winter of 2017 for all katabatic wind events. Events were identified using data from an automatic weather station located on the nearby Nansen Ice Shelf and data from the moored sensors themselves. Satellite data were used to calculate the area of the polynya. They then develop a parameterization from a combination of weather station wind data, reanalysis data, and satellite derived polynya area data to calculate a high salinity shelf water (HSSW) production time series for TNB over an extended period in order to place the results from the mooring in a temporal context. The authors show that the variability in wind speed prior to the HSSW production events plays a significant role in HSSW production, and that the role changes in different parts of the winter. They also find their parameterized result for 2017 compares well with the result from the mooring data, lending credibility to the multiyear time series. Their derived time series finds an increasing TNB HSSW production rate since 2015.

The ambition of improving our ability to obtain accurate estimates of HSSW production without the need for moored instruments is, I believe, a worthy one. In the main, my comments cover some areas where the choice of methodology needs more evidence in its support.

The text is nicely written, with few typographical errors that I spotted. There is an occasional curious word choice. I have made some minor textual suggestions in the attached Word document, in tracked changes. A general language point is that the word following a colon should not start with a capital letter, and for some unknown (to me) reason, ice features do not carry a definite article, so “the Nansen Ice Shelf” for example should be simply “Nansen Ice Shelf”.

I do have some more significant concerns that I would like to see addressed before this manuscript was published. I outline those below, and they are also noted in brief in the Word document.

The study has two quite separate parts: analysis of the moored sensor data, and the calculation of the HSSW production rate time series using AWS, remotely sensed, and model data. The principal connection is that the result of the mooring analysis is used to provide a degree of validation for the parameterization. It seems a shame that one of the key results from the mooring analysis (for me at least), that the HSSW productivity of a katabatic event has a significant dependence on the preceding wind variability, is not used in the parameterization. It might be that for some reason this is too difficult to implement, but it seems like a missed opportunity. It would be useful to mention that this is not being attempted here, and why that is the case. Perhaps the authors plan to implement such a scheme in the future.

We agree that there are two somewhat distinct parts of the manuscript and thank you for raising the point that inclusion of katabatic wind characteristics in the heat flux parameterization is an important way to tie the two together. We argue that dependence on preceding katabatic wind conditions is actually already implicitly built into the existing parameterization, however, through its dependence on polynya area (Equation 12). As we discuss in Lines 124-135, polynya area is highly dependent on katabatic wind event frequency and duration. Prior studies have highlighted the close connection between polynya area variability and the katabatic wind regime (e.g., Ciappa et al., 2012; <https://doi.org/10.1016/j.rse.2011.12.017>), with Aulicino et al. (2018; <https://doi.org/10.3390/rs10030366>)

specifically finding interannual variability in polynya area to be dependent on katabatic wind frequency and duration. We have added text highlighting the connection between our initial analysis of dependence of HSSW production rate/polynya area on katabatic wind characteristics and the parameterization of HSSW production rate from heat flux to Lines 232-235. We believe you have raised an insightful point and hope that the manuscript now sufficiently addresses it.

The salinity and temperature data in Figure 2 (and also the example event in Figure 3) raise some important questions. Had the authors not indicated that the sensors were carefully calibrated, they would have appeared to be a bit suspect. The temperature of the majority of what is classed as HSSW is, in fact, below the surface freezing point and therefore formally ice shelf water (ISW). I also wonder if the authors can check the location of the freezing point line in Figure 2 – it looks a shade warm to me, possibly by about 0.02 degrees C? That would help bring more of the high salinity water close to the freezing point, or at it. Even then, much of the high salinity water is below even the in situ freezing point (ie taking into account the pressure at 47 m depth), which is odd and needs further comment. Having in situ supercooled water (if that's what it is), masquerading as HSSW, is worth a mention.

It is not unusual for HSSW observed in Terra Nova Bay to fall partly or entirely below the surface freezing point, as observed in previously published TS diagrams from Terra Nova Bay:

- Figure 2 in Orsi and Wiederwohl (2009; DOI: [10.1016/j.dsr2.2008.10.033](https://doi.org/10.1016/j.dsr2.2008.10.033)),
- Figure 6 in Yoon et al. (2020; DOI: [10.5194/os-16-373-2020](https://doi.org/10.5194/os-16-373-2020))
- Figure 2 in Le Bel et al. (2021; DOI: [10.1017/S0954102021000146](https://doi.org/10.1017/S0954102021000146))
- Figure 7 in Stevens et al. (2017; <https://doi.org/10.1017/aog.2017.4>)

Oshima et al. (2022) <https://doi.org/10.1126/sciadv.adc9174>) also notes the presence of supercooled HSSW in the Cape Darnley Polynya.

The defining temperature of HSSW is often described as “near freezing point”, but not necessarily above it. However, we agree that its apparent supercooling, especially relative to in-situ freezing point, requires some further thought. We believe there are two mechanisms that result in supercooled HSSW. The first is supercooling of the surface water via intense winds, which has been documented in both Arctic (Skogset et al., 2009; DOI: [10.3189/002214309788608840](https://doi.org/10.3189/002214309788608840)) and Antarctic polynyas (Thompson et al., 2020; DOI: [10.5194/tc-14-3329-2020](https://doi.org/10.5194/tc-14-3329-2020)). In the case of the latter, CTD casts

show most if not all of the water column to be supercooled relative to the surface freezing point, and supercooled water relative to in situ freezing point is evident as deep as 20m (resulting in frazil ice formation). The second mechanism is by mixing with ISW, which in TNB will have an endmember temperature around freezing point at ~250m, the grounding line depth of Nansen Ice Shelf. It is possible that ISW at depth is entrained by the deepening mixed layer in autumn months or injected into the fully

homogenized water column in winter months via circulation under Nansen Ice Shelf around ~250m, and is then mixed up to the depth of our instrument at 47m. This would help to explain the presence of water that is supercooled not only relative to surface freezing point, but to *in situ* (47m) freezing point as well.

ISW is defined by its temperature below surface freezing point and by its mechanism of formation via the mixing of HSSW with meltwater. Because ISW is formed *from* HSSW, they are distinguished in temperature-salinity space by salinity (or equivalently, because of the cold temperatures, by density), even in the case where HSSW is also below surface freezing point. For example, Yoon et al. (2000) distinguishes between ISW and HSSW using a threshold of density = 1028 kg/m^3 . Therefore, we respectfully disagree that the HSSW observed in our TS diagram is technically ISW because it is below the surface freezing point.

We have double checked our calculation of the freezing point in Figure 2 and believe it to be calculated correctly. It is calculated using the Gibbs SeaWater software package as:

```
gsw.t_freezing(absolute_salinity,0,0)
```

Where absolute salinity is converted from practical salinity using the function `gsw.SA_from_SP(salinity,0,lon,lat)`, the depth is set to 0, and the saturation fraction is set to 0 (setting the saturation fraction to 1 makes virtually no difference).

When HSSW is formed at the surface, it will be at the surface freezing point, and so can only be cooled further by mixing with ISW, or by coming into contact with deep ice. I assume the authors are assuming that the former case applies here, which implies quite a lot of recycling of ISW within the polynya. If much of that ISW is recycled into HSSW the calculation of the fraction of HSSW converted to ISW (line 139 onwards) is likely to be less relevant. I think the manuscript would be improved if the authors discussed this issue. The fact that the majority of the HSSW detected is formally ISW needs a mention at least. It would be useful to include the T-S from a CTD profile, to give the data in Figure 2 some context. I recognize that they would be a summer snapshot only, but it would help the reader get their T-S bearings.

As discussed above, supercooling of HSSW relative to the surface freezing point is likely caused by katabatic wind-induced heat loss rather than mixing with ISW, and given that ISW and HSSW are distinguished by a density threshold of 1028 kg/m^3 rather than temperature (Yoon et al., 2020), we disagree that the majority of HSSW observed here is ISW/mixed with ISW. However, we acknowledge that the HSSW below *in situ* freezing point could be indicative of mixing with ISW, and that you raise a pertinent point about ISW recycling. Though, we do not believe that ISW recycling would affect our budget calculation for two reasons: The first is that whether or not ISW is recycled does not change the $\sim 0.48 Sv$ (was previously $0.6 Sv$ but has since been amended; Please see Page 1 of this document for details) production rate that we obtain from the moored salinity time series nor the $0.4 Sv$ export rate measured by Gordon et al. (2009). That is, $0.48 Sv$ still need to enter Terra Nova Bay and $0.4 Sv$ need to exit. What would change is the source of water that is salinified and converted into HSSW. For example, if we assume ISW is not recycled, then $0.48 Sv$ of modified Circumpolar Deep Water (mCDW, the dominant ambient water mass in Terra Nova Bay) is salinified by brine rejection and converted to HSSW. If we assume ISW is recycled, then we might expect something like $0.44 Sv$ of mCDW and $0.04 Sv$ of ISW converted into HSSW. Please see our rough box model diagram below detailing this thought

process. Secondly, instances where the temperature measured by the 47m instrument drop below in situ freezing point are episodic and relatively brief (see below figure), so while they may indicate mixing with ISW, these occurrences are sparse and therefore arguably insignificant to our present analysis, though a point of interest for future study.

Though we believe our TS diagram (Figure 2) to be fairly typical compared to others in TNB (as discussed on Page 6 of this document), we understand that context for our TS diagram (Figure 2) would be helpful. A copy of Figure 2 is shown below overlain with red dots corresponding to the downcast TS data from the CTD cast taken upon deployment of the mooring. This is to illustrate the TS space occupied by the TS diagram and is not how we would intend to show the data in a final figure. Because of the restricted TS space of the summertime snapshot relative to the whole year of moored data, we are not sure it provides useful context either as part of Figure 2 or as a separate figure. If you disagree, we will be happy to work further on a version of Figure 2 that incorporates the CTD data, or alternatively, could consider incorporating TS data from deeper instruments on the mooring, though we prefer to leave that data for a later publication.

I have a difficulty with the calculation of the HSSW production. First, equation (1) is expressed as an integral over the upper 47 m (zero to H) of the water column, except that the authors only have data from 47-m depth. It would be less confusing to express it as $H \cdot S \cdot \rho$. Eqn (1) gives the mass of salt between the surface and 47 m, and (2) shows how that salt mass is increasing during the event. Eqn (3) then integrates across the area of the polynya, and (4) converts this salt gain to an HSSW production rate. What causes me concern is that I don't understand the basis for the assumption that the increase in salinity happens only in the water column between the instrument and the surface. Is there any evidence to suggest that it doesn't descend to the sea floor? This is what I would have expected, but it's difficult to judge without knowledge of the stratification. Overall, my suspicion is that the answers from this calculation might well form a good indicator of HSSW production and its relative intensity during the events, but not an absolute value.

Your concern with this value is well taken, as the lead author debated over the appropriate depth when first working through this calculation. We thank you for raising the question of proper integration depth, as it has caused us to reconsider our choice of 47 m.

You are correct that the brine descends past the 47 m instrument and that we see salinity increases across the full water column, including the deepest instrument at 360 m. However, the correct depth of integration should be the depth of the in-situ frazil ice formation layer (the upper tens of meters of the water column), not the entire depth of the water column. This is because we are interested in the salt flux directly caused by in-situ ice formation. If we were to integrate across the salinities from instruments at greater depths, the calculated salt flux would represent both the flux due directly to frazil ice formation/brine rejection at the surface, as well as the flux of brine downward as it sinks/is mixed to depth. Put another way, the flux of salt entering the system via in-situ ice production, plus the flux of salt traveling through the water column. To estimate brine rejection, we should only consider the flux due to in-situ ice formation; integrating across all available depths would include the flux due to mixing/sinking and would therefore result in an overestimation of HSSW production rate. To confirm this, we recomputed the salt flux term (dm/dt) using the full set of moored salinity measurements integrated across depths of 47m - 360m. This resulted in an average HSSW production rate of 3 Sv, which is unrealistically high compared to the measured 0.4 Sv exported from the region by Gordon et al. (2009) and estimates of production rate using net heat fluxes, both our own and that of Fusco et al. (2009) (~1 Sv).

That said, while the depth of the "frazil ice formation layer" is the appropriate depth of integration, we no longer believe 47m is the correct depth. For frazil ice formation to occur, the water must be below in-situ freezing point. CTD casts in Thompson et al. (2020; [DOI: 10.5194/tc-14-3329-2020](https://doi.org/10.5194/tc-14-3329-2020)) show the presence of supercooled water relative to in-situ freezing point down to about 20 m water depth, and a latent heat anomaly indicative of frazil ice formation above this depth. In their calculation of TNB ice production rates from salinity anomalies, they set their depth of integration to values around 20 m, based on the depth of in-situ freezing. Assuming the salinity of water is well mixed and uniform from the sensor at 47m to the surface, calculation of in situ freezing point at our mooring site matches up with the measured in-situ temperature when a depth of 30 m is used, meaning the bottom of the "frazil ice formation layer" is 30 m. Therefore, we have redone our calculations using an integration depth of 30 m instead of 47 m. This changes the average production rate from ~0.6 Sv to ~0.4 Sv.

The authors use the assumption that the HSSW is produced only during katabatic events, and the

definition of those events is highly prescriptive. I haven't tried to calculate the fraction of the period between July and October when katabatic events were active, but unless that is a very large fraction, some sort of argument is needed to convince the reader that HSSW being produced between katabatic events is insignificant. For example, if events occupied only 10% of the time then the polynya during non-katabatic conditions would need to be a very weak produced of HSSW to be making an insignificant contribution to the total.

We apologize that we are not confident we understand your concern about HSSW being produced in between katabatic wind events. Please let us know if the following does not address your concern.

We do not assume that HSSW production only occurs during katabatic wind events, rather, we:

- 1) First, identify periods of time during which the density timeseries measured by the instrument exceeds 1028 kg/m³, i.e. that HSSW is present
- 2) Divide these time periods into distinct "HSSW production events", defined to start when density at the instrument first increases above 1028 kg/m³ and defined to end when density falls below 1028 kg/m³ for more than an hour.
- 3) Filter out events that are not definitively associated with the opening of the polynya, i.e., proceed/are concurrent to katabatic winds, proceed a transition in the acoustic surface tracking from hard to soft reflectors at the surface, and are concurrent to eastward flowing surface currents
- 4) Use the increase in salinity (e.g. Figure 3c, copied below) associated with the start of each of the "verified" events to estimate brine rejection rates, from which we estimate HSSW production rates.

Because we are capturing every instance in the density time series where density increases from below 1028 kg/m³ to above it, there should not be any HSSW production in between defined events, except for those few we discard because they do not meet the AST and current velocity criteria. We realize that our description of how HSSW production events were detected and formally defined is not very clear in the manuscript, and have updated Lines 342-370 to better describe the methodology.

ISW meltwater fraction estimate in line 373-380. This calculation needs to be re-written. T_meltwater and S_meltwater have been assigned the same values as T_ISW and S_ISW. Presumably a typo. S_meltwater should be zero, and T_meltwater should be roughly $T_{HSSW} - L/c$, where L is the latent heat of fusion of ice, and c is the specific heat capacity of water. This is about 86 degrees C. This then gives about the correct answers.

Thank you for catching this. Please see the corrected values, now on Line 413.

Lines 167+ This doesn't seem right. They are each expressed as an average for an entire year. The authors' value is also for an entire year and should not be reduced in this way. The adjustment would have been correct had they been expressed in terms of production rate per month. That would place the observed value in the middle of the published values.

This is something the we had debated on, and we appreciate that you took the time to address it. We think you are correct.

We have removed this adjustment and updated the text in which we compare our average value to previously published values accordingly (Line 172 onward).

Line 196 Can the authors check this. I'm not sure the correction can have been multiplicative – the units wouldn't work out. (Or am I missing something?)

Thank you very much for catching this and bringing it to our attention - you are correct that the units mean the correction cannot be multiplicative. We have also checked our conversion of the reported $0.15 * 10^2 \frac{m^3}{day}$ into Sv and realized that we made a large error. The value is not 173 Sv, but rather:

$$0.15 * 10^2 \frac{m^3}{day} * \frac{1 day}{86400 s} * \frac{1 Sv}{10^6 m^3 s^{-1}} = 1.74 * 10^{-9} Sv$$

Fusco et al (2009) states that the correction is "applied", and no further detail is given on how. Given the units we would assume the correction is additive, but it is such a small value that it seems that it would be insignificant. Because we are unsure of how it is applied, we have removed mention of the correction from our manuscript.

Lines 348-350 I think this is the first mention of other sensors, so they come as a surprise at this stage. Might be worth giving a fuller description of the mooring earlier in the manuscript, and just saying that only the uppermost instruments will be used here (and perhaps say why).

We have added text to Lines 275-279 that mentions the full mooring and explains why only the shallowest sensors are used in the present manuscript. We have also added more detail on the ADCP in response to your in-document comments (Lines 281-287).

There are several other comments in the attached marked-up version for the authors to review. I have not attempted to review the calculation of the surface heat fluxes – I leave that to a reviewer who knows more about that than me (I've also run out of time – apologies).

REVIEWER COMMENTS

Reviewer #3 (Remarks to the Author):

Second review of "High Salinity Shelf Water production in Terra Nova Bay, Ross Sea from high-resolution near-surface salinity observations", by Miller et al.

Many thanks to the authors for their detailed response to my comments. I've learnt a lot from them, not least that in situ supercooling due to heat loss to the atmosphere can be detected at many tens of metres depth. We don't (yet) have similar observations in the Weddell Sea, and my education was clearly lacking. I also now see the need for the definition of Ice Shelf Water to be modified to take that into account.

I clearly got the wrong end of the stick on one or two points, but having given the study some more thought, I think there remain two areas that I would like to understand properly. Although I might not be the brightest, I'm probably not the dimmest of the likely readership; if I don't understand the arguments, then I imagine others might struggle too. My revised opinion is that, while the calculation of ice production from the AWS and polynya area data seems fine to me, the calculation of HSSW production from the mooring data needs to be revised before publication.

I have returned a marked-up pdf. This covers some (very) minor suggestions for the text, and some more significant questions and comments. On the whole, I found the manuscript to be very well and clearly written.

My two major comments are:

1. My understanding is that the rate of increase in salinity at 47 m depth is calculated for the period of the increase in salinity resulting from a katabatic wind event. This is then converted into an HSSW production rate for the entire polynya area for that interval in time. That figure typically comes out as a few tenths of a Sv. The 28 production rates, one for each of the 28 events, are then averaged to get a mean HSSW production rate. But this is the rate of production during the individual events. The authors' view is that no HSSW is being produced between these events. I think, therefore, that the mean production rate for the entire year should be that mean value, multiplied by the total time interval of production (eg 28 x mean HSSW production period) divided by one year. Put another way, HSSW production, integrated across all the production events, needs to be divided by one year in order to get an annual production rate. The authors' approach seems to suggest that the production rate during an individual event is sustained over the entire year. However, the rate of production (in Sv) must be averaged over a time interval, and it would normally be one year. One could divide the integrated production by the interval between July and October, and say that X Sv is being generated during that winter production period. Either that or I'm missing something...

I recognise that this would dramatically reduce the calculated value, but see below.

2. The authors responded to my question about the depth over which the salinity increase needs to be integrated. They told me that they wished to determine the salinity increase due to production of frazil ice within the water column, and referred to the paper by Thompson et al. But note that Thompson et al were investigating the interesting process of frazil production in the upper few tens of metres of the water column; they were not looking at HSSW production per se. I have also always assumed that the vast majority of sea-ice production takes place at the sea surface, with brine rejection causing overturning. In the case of strong katabatics, wind-induced mixing is presumably the principal agent causing supercooled water to penetrate into the upper few tens of metres, assisted by a negative buoyancy gradient.

I still don't understand how the authors' approach can be correct for calculating the total HSSW production during each production event using the salinity increase over the upper 30 m or so. Whether my assumptions about where the majority of sea ice is formed are correct or not, the point is that the aim must be to calculate a salt budget, which must include the entire water column not just the component due to frazil formation in the upper few tens of metres. The authors seem to think that calculating the salt budget over the entire water column is in some way double-counting, but I don't think that's correct. In the one-dimensional view enforced by having data from a single mooring, the change in vertically-integrated salt content must surely represent change due to sea-ice production, which is what is needed. As the authors point out in their response to this comment, this would give a value for HSSW production an order of magnitude greater than that observed. However, as pointed out above, this is for a short period only, namely, the duration of the production event.

We sincerely thank Dr. Nicholls for his continued time and effort in reviewing our manuscript. After considering his latest comments carefully, we agree that brine rejection should be computed from the integral of full water column salinity, rather than salinity only at the shallowest instrument, and that values should be converted to an annual average. Please find our response to Dr. Nicholls' comments below, along with the following description of other notable changes made:

- For clarity, parts of the final paragraph of the introductory text and the first paragraph of the “variability tied to katabatic winds” section were moved into a new section titled “HSSW production rates from moored salinity time series” (Lines 96-129)
- Because it may be of interest to the Ross Sea community, we added an estimate of ice production rate based on the measured brine rejection rate from the mooring to Lines 126-129. Ice production rates for each HSSW production event are now given in Supplementary Table 1. Their calculation is described in the Methods section, Lines 431-435
- In light of a recent study of shipboard bulk and integral heat flux measurements in TNB that describes a heat transfer coefficient of 2.2×10^{-3} (Guest et al., 2022b; 10.1029/2021JD034904) we have expanded the coefficient range used to bound our heat flux calculations to include this higher coefficient. This is referenced in Line 222, discussed in Lines and 537-542, and reflected in an updated Figure 5.
- Because the use of the 2.2×10^3 coefficient results in much larger heat fluxes, and therefore heat-flux-based HSSW rate estimates, we opt to return to our original assignment of S_{HSSW} and S_{LSSW} in Equation 4 to their respective average values observed across the 27 production events in 2017 in the moored salinity data (Lines 556 – 557), which brings the resulting production rates, both using the upper and lower bound heat flux coefficients, into a more reasonable range of our mooring-based estimate. Note these average values are slightly different from those reported in the original submission as we now utilize salinity across the full water column rather than just at the 47 m sensor. As Dr. Nicholls has rightly pointed out, the sensitivity of the heat-flux-based estimates of HSSW production rate to the choice of $(S_{HSSW} - S_{LSSW})$ is a large source of uncertainty, which we now discuss in Lines 557 – 561.
- In the first round of revisions, we removed the mention of Fusco et al. (2009)'s use of reanalysis wind speeds and associated application of a correction factor of $0.15 \times 10^2 \text{ m}^3 \text{ day}^{-1}$ because Dr. Nicholls pointed out that we had wrongly assumed this factor was multiplicative and the original source text does not describe how exactly this correction factor applied. We have added parts of this discussion back into the present manuscript (Lines 210-215) because the lack of katabatic wind dynamics in reanalysis is an important source of uncertainty to acknowledge, while being careful not to mischaracterize the correction factor.
- Because we now use salinity across all 7 sensors instead of just the shallowest sensor, we updated the criteria for HSSW production events to include a formal metric for a mixed water column (lines 97-101). This resulted in the disqualification of one production event in June, bringing our count from 28 to 27 events.

Reviewer #3 (Remarks to the Author):

Second review of “High Salinity Shelf Water production in Terra Nova Bay, Ross Sea from high-resolution

near-surface salinity observations”, by Miller et al.

Many thanks to the authors for their detailed response to my comments. I’ve learnt a lot from them, not least that in situ supercooling due to heat loss to the atmosphere can be detected at many tens of metres depth. We don’t (yet) have similar observations in the Weddell Sea, and my education was clearly lacking. I also now see the need for the definition of Ice Shelf Water to be modified to take that into account.

I clearly got the wrong end of the stick on one or two points, but having given the study some more thought, I think there remain two areas that I would like to understand properly. Although I might not be the brightest, I’m probably not the dimmest of the likely readership; if I don’t understand the arguments, then I imagine others might struggle too. My revised opinion is that, while the calculation of ice production from the AWS and polynya area data seems fine to me, the calculation of HSSW production from the mooring data needs to be revised before publication.

I have returned a marked-up pdf. This covers some (very) minor suggestions for the text, and some more significant questions and comments. On the whole, I found the manuscript to be very well and clearly written.

My two major comments are:

1. My understanding is that the rate of increase in salinity at 47 m depth is calculated for the period of the increase in salinity resulting from a katabatic wind event. This is then converted into an HSSW production rate for the entire polynya area for that interval in time. That figure typically comes out as a few tenths of a Sv. The 28 production rates, one for each of the 28 events, are then averaged to get a mean HSSW production rate. But this is the rate of production during the individual events. The authors’ view is that no HSSW is being produced between these events. I think, therefore, that the mean production rate for the entire year should be that mean value, multiplied by the total time interval of production (eg 28 x mean HSSW production period) divided by one year. Put another way, HSSW production, integrated across all the production events, needs to be divided by one year in order to get an annual production rate. The authors’ approach seems to suggest that the production rate during an individual event is sustained over the entire year. However, the rate of production (in Sv) must be averaged over a time interval, and it would normally be one year. One could divide the integrated production by the interval between July and October, and say that X Sv is being generated during that winter production period. Either that or I’m missing something...

We understand your point, as we had originally debated whether to convert our event-average estimate to an annual average by multiplying by the ratio of the number of months out of active HSSW production to the total number of months in a year (4/12). This was based on a similar rationale employed in Gordon et al. (2009), who used 6/12.

A more refined conversion could be as you suggested: a ratio of the total time interval of our observed production events to a year. However, our criteria (Lines 105 – 116) resulted in our removal of a number of events that likely were representative of HSSW production but were obscured by advective anomalies

(perhaps eddies). Therefore, the sum of time captured by our identified HSSW production events would be an underestimation of the total time of HSSW production. The alternative that we propose is the sum total of time across the months of active HSSW production, July-October, during which wind speeds at AWS Manuela exceed 25 m s^{-1} , our defining threshold for katabatic winds. This is 42.11 days, which results in a conversion factor of $42.11/365 = 0.115$ (Lines 426-430). Our rationale is that during the HSSW production months of July-October, whenever katabatic winds blow, the polynya will open, the surface will freeze, new ice will form, and HSSW will be produced. An analysis of AST returns as a function of wind speeds during this period suggest the polynya to open whenever wind speeds exceed 25 m/s. Here, the standard deviation of AST pings is calculated over 10-minute bins, with higher standard deviations indicating the polynya is open (pings are highly scattered when the polynya is open due to reflection off of frazil ice, injected air bubbles, and sea spray, as discussed in Lines 108-111 of the manuscript and shown in Figure 4b) and low standard deviations indicating the polynya is closed.

One could argue for the conversion factor to be calculated based on a wind speed threshold of 20 m s^{-1} . This would be $68.56/365 = 0.188$, resulting in an annual-average production rate of 0.69 Sv rather than the 0.43 Sv that we report. However, the above figure shows 25 m s^{-1} to be more clear-cut wind speed threshold, with virtually all standard deviations of 10-minute AST bins exceeding 2 m. The 20 m s^{-1} threshold would likely include periods of time when HSSW production is not or only partially occurring, such as when the polynya is first opening and as it is closing

Because we convert the mooring-based estimate into an annual average, we must do the same for the heat-flux-based estimates. This is done using the simple months-of-active-HSSW-formation ratio employed in Gordon et al. (2009) and described above, 4/12. (Lines 561 - 563).

I recognise that this would dramatically reduce the calculated value, but see below.

2. The authors responded to my question about the depth over which the salinity increase needs to be integrated. They told me that they wished to determine the salinity increase due to production of frazil ice within the water column, and referred to the paper by Thompson et al. But note that Thompson et al

were investigating the interesting process of frazil production in the upper few tens of metres of the water column; they were not looking at HSSW production per se. I have also always assumed that the vast majority of sea-ice production takes place at the sea surface, with brine rejection causing overturning. In the case of strong katabatics, wind-induced mixing is presumably the principal agent causing supercooled water to penetrate into the upper few tens of metres, assisted by a negative buoyancy gradient.

I still don't understand how the authors' approach can be correct for calculating the total HSSW production during each production event using the salinity increase over the upper 30 m or so. Whether my assumptions about where the majority of sea ice is formed are correct or not, the point is that the aim must be to calculate a salt budget, which must include the entire water column not just the component due to frazil formation in the upper few tens of metres. The authors seem to think that calculating the salt budget over the entire water column is in some way double-counting, but I don't think that's correct. In the one-dimensional view enforced by having data from a single mooring, the change in vertically-integrated salt content must surely represent change due to sea-ice production, which is what is needed. As the authors point out in their response to this comment, this would give a value for HSSW production an order of magnitude greater than that observed. However, as pointed out above, this is for a short period only, namely, the duration of the production event.

We have thought through this carefully and have come to agree that the salt content integrated across the full mooring line is necessary. We realize that our previous assertion that salinity at 30m ("frazil ice depth") can be assumed identical at 47m relies on the assumption that the water column above 47m is well mixed. Well, full water column destratification is a requirement for HSSW production and if we assume the near surface is well mixed, then we must assume the whole water column is well mixed if HSSW is being produced. Therefore, rejected brine would be distributed throughout the water column and the full set of MicroCATs (which reach the seafloor) must therefore be used. We are grateful to you for continuing to question the depth of integration and feel our methods are now more rigorous and take fuller advantage of the mooring's uniquely high density of salinity measurements in the upper ocean.

You are right that converting to an annual average brings the large HSSW production rates obtained from integrating across the full water column depth to a more typical range. In fact, the average value remains $\sim 0.4 Sv$, which is what we had reported in the last round of revisions.

Comments from the PDF of the manuscript

I don't quite follow why the placement on a bathymetric high gives better access to the upper 50 meters. If deployed deeper, you could use a longer mooring line? What am I missing?

We acknowledge that our assertion here was vague. Now that we are using the full set of MicroCATs across depth, we assert that the abundance of instrumentation and the relatively shallow location on the bathymetric high result in an unusually high density of instruments across the full water column (Line 84-87). We hope the value of the placement on a bathymetric high is now clearer.

This sentence needs a bit of work. Maybe delete "despite", and insert "more" between "was" and "likely". And "proceed" isn't really the opposite of "precede". Perhaps "follow" instead? (All assuming I understand the intended meaning.)

Thank you for pointing out this typo. We've reworked this sentence from

"For example, one event occurring in late June was removed from further analysis despite because it was found to immediately proceed a sudden reversal of strong Northward currents, indicating that the observed increase in salinity was likely associated with advection than with in-situ ice production."

into

"Events were discarded if salinity increases appeared to be associated with processes other than in-situ ice production, such as advection by eddies.." (Lines 104-106)

And

"some events were discarded because of strong shoreward currents or abrupt changes in current direction that were clearly decoupled from the westerly katabatic winds and possibly associated with sub-mesoscale eddy activity previously observed in TNB^{36,37}. " (Lines 115 – 117)

assessed" sounds a bit like you are validating it (the word is ambiguous). Would "determined" be more precise, since you actually validate it using the mooring based estimate?

This has been changed from

"...interannual variability of HSSW production rates in TNB. The latter is **assessed** using a modified approach to a previously published method for estimating HSSW production rates from parametrized net heat fluxes in TNB^{29,31,32}, which we first validate against the average of our mooring-based estimates in 2017..."

to

"We use these estimates to explore several aspects of HSSW production, including the conversion of HSSW into Ice Shelf Water (ISW) and its variability in relation to the katabatic wind regime, as well as to ground truth a method for estimating production rates from parameterized net heat fluxes in order to **examine** potential interannual variability in the context of broader Ross Sea salinity trends." (Lines 92-96).

Just a comment - I much prefer this approach to the one above. I think a full uncertainty analysis of the first approach would show the the TISW volume is statistically indistinguishable from zero. Without such an analysis, of course, I can't judge properly.

We now give the estimated ISW production rate in terms of order of magnitude ($\mathcal{O}(0.1Sv)Sv$) rather than to the decimal (e.g., 0.08 Sv), given the large uncertainties associated with this exercise. As a side note on this section, now that our estimates of HSSW production are converted to annual averages, we now use the annual-average value of 0.2 Sv, rather than the seasonal values of 0.4 Sv, reported by Gordon et. al (2009) for this budget analysis.

Is there anything to say that this is going to be constant from year to year? Presumably not, although it's clearly an improvement on the previous approach. Identifying the way the relevant interval might change from year to year is one good motivation for continuing mooring deployments.

Mooring data in TNB (CLIMA mooring "D", further offshore of our mooring) from 1995-2008 show a reoccurring increase in salinity along the water column each year between July and November, which Rusciano et al (2013) attributes to HSSW formation. This is consistent with our identified HSSW production interval. We therefore expect this time period to remain fairly consistent year-to-year, but your point about potential changes in the future as another source of uncertainty is well taken.

Continuing on the discussion from the first submission, the authors say above that the method they have developed works, so removing the need for costly moorings, but then go on to say we need more costly moorings to prove that the method works. Can't really have it both ways. Perhaps say that this pilot experiment shows that the method holds promise for removing the need for moorings, but more validation is needed, or explain why more validation is needed despite saying above that the method works.

This is a very fair point, thank you. We have reworked this portion of the Implications section to explicitly identify and discuss two aspects of our work that further mooring deployments would help address. We have also somewhat deemphasized the vague need for validation of the heat flux method, and have now rather pointed out that the strong dependence on HSSW variability on polynya area observed in both our salinity- and heat-flux-based estimates could mean that the former could be inferred from the latter, a relatively easily obtained variable.

This is discussed by the authors in the response letter. As this is likely to make a major contribution to the uncertainty, enumerating its contribution is important. Without having the years of mooring data that the authors say are needed for improving estimates, it is difficult to get a handle on the likely variability of ΔS , but some assessment of its effect would be helpful.

The sensitivity of the calculation of production rates from net heat fluxes to the ΔS term has become increasingly clear to us throughout this revision process and we agree that its important for the reader to have a sense for the magnitude of this sensitivity. This is now discussed in Lines 558-562.

REVIEWERS' COMMENTS

Reviewer #3 (Remarks to the Author):

I am grateful to the authors for their full response to my comments and concerns. Their explanation of the temporal extrapolation of the HSSW-production intervals seen in the mooring data across the entire season is very clear. My only comment is that I find the explanation to me rather clearer than the explanation in the manuscript. But I accept that might be my problem.

I have made a very few, very minor comments on the pdf attached, which can be ignored at will.

Good luck with the remainder of the production process.

Our sincerest thanks to Dr. Nicholls for his time, effort, and interest in improving the quality of our manuscript.

Please find the reviewer comments in black and the authors' responses in yellow, below.

Line 31

I suppose I'm a bit of a Wunsch-ist, and see the GTC driven by winds, and substantially modified by deep water formation. But bearing in mind Arnold G is on the author list, if he is happy with "drives" rather than "contributes to", for example, then I guess I should be too.

We appreciate that there are different ways to define GTC and its "driving" forces. We have changed our wording in the manuscript to:

"... (AABW), a water mass that constitutes the bottom limb of meridional overturning circulation and directly impacts the heat, carbon, oxygen, and nutrient inventories of the deep ocean¹⁻³."

Line 113

Can we define this acronym here? Not used previously. I understand it's defined in Methods below.

The acronym is now defined.

Line 268

Similar comment to earlier. A Wunsch-ist would probably claim that the strength is determined by global winds. But I'm not expert in these matters.

A fair point – it would have perhaps been better for us to specify "...the strength of the *bottom limb* of the MOC". We have changed this line to:

"Variability in the properties and production rate of AABW directly impacts the capacity of the ocean to sequester atmospheric heat and carbon and replenish abyssal oxygen and nutrient inventories^{60,61}."

Line 295

"key"? Or is "various" what's meant? Perhaps "a spread of locations"?

The last few sentences of the implications section have been reworked and our call for additional moorings has been fleshed out. We now separately call for additional moorings to a) assess spatial variability in HSSW production rates caused by down-fetch heat loss within TNB and b) assess how variability in TNB HSSW production propagates downstream in the Drygalski Trough.